# 21st-century modeled permafrost carbon emissions accelerated by abrupt thaw beneath lakes

Katey Walter Anthony [1], Thomas Schneider von Deimling[2,3], Ingmar Nitze [3,4], Steve Frolking[5], Abraham Emond[6], Ronald Daanen[6], Peter Anthony[1], Prajna Lindgren[7], Benjamin Jones[1] & Guido Grosse [3,8]

Permafrost carbon feedback (PCF) modeling has focused on gradual thaw of near-surface permafrost leading to enhanced carbon dioxide and methane emissions that accelerate global climate warming. These state-of-the-art land models have yet to incorporate deeper, abrupt thaw in the PCF. Here we use model data, supported by field observations, radiocarbon dating, and remote sensing, to show that methane and carbon dioxide emissions from abrupt thaw beneath thermokarst lakes will more than double radiative forcing from circumpolar permafrost-soil carbon fluxes this century. Abrupt thaw lake emissions are similar under moderate and high representative concentration pathways (RCP4.5 and RCP8.5), but their relative contribution to the PCF is much larger under the moderate warming scenario. Abrupt thaw accelerates mobilization of deeply frozen, ancient carbon, increasing $^{14}$C-depleted permafrost soil carbon emissions by ~125–190% compared to gradual thaw alone. These findings demonstrate the need to incorporate abrupt thaw processes in earth system models for more comprehensive projection of the PCF this century.

[1] Water and Environmental Research Center, University of Alaska Fairbanks, Fairbanks, AK 99775, USA. [2] Max Planck Institute for Meteorology, 20146 Hamburg, Germany. [3] Alfred Wegener Institute Helmholtz Centre for Polar and Marine Research, 14473 Potsdam, Germany. [4] Institute of Geography, University of Potsdam, 14476 Potsdam, Germany. [5] Institute for the Study of Earth, Oceans, and Space, University of New Hampshire, Durham, NH 03824, USA. [6] Alaska Division of Geological & Geophysical Surveys, Fairbanks, AK 99775, USA. [7] Geophysical Institute, University of Alaska Fairbanks, Fairbanks, AK 99775, USA. [8] Institute of Earth and Environmental Sciences, University of Potsdam, 14476 Potsdam, Germany. Correspondence and requests for materials should be addressed to K.W.A. (email: kmwalteranthony@alaska.edu)

Northern permafrost soils represent the largest terrestrial organic carbon pool (1330–1580 petagrams, Pg)[1] on Earth. While frozen, this soil carbon reservoir is stable. However, recent observations[2–5] and projections[1,6–10] of future soil warming and permafrost thaw suggest that permafrost soil carbon will be increasingly vulnerable to decomposition by microbes that generate the greenhouse gases carbon dioxide ($CO_2$) and methane ($CH_4$). This release of permafrost carbon as greenhouse gases constitutes a positive feedback likely to amplify climate warming beyond most current earth system model projections[1].

Recent permafrost carbon feedback (PCF) estimates indicate 23–174 Pg of permafrost soil carbon emission by 2100 under the Representative Concentration Pathway (RCP) 8.5 scenario[1,7,8] and 6–33 Pg C under RCP4.5[7,8]. Such permafrost carbon emission trajectories, which were not included in the Intergovernmental Panel on Climate Change (IPCC)'s Fifth Assessment Report (AR5)[11], would increase global temperatures predicted for 2100 by ~$0.3 \pm 0.2$ °C (RCP8.5)[6]. Compared to $CO_2$, $CH_4$ is considered the lesser cause of warming, responsible for < 20% of the total PCF temperature increase[6] despite its much higher global warming potential[12]. The land-surface carbon models, which operate at large landscape scales and account for very basic permafrost-soil thermodynamics and its control on soil organic carbon decomposition[6–10], represent only gradual, top-down thaw of permafrost soils by active layer deepening (i.e., increasing seasonal surface thaw). They do not consider the mechanism of deeper, abrupt thaw.

Thermokarst, the most widespread form of abrupt permafrost thaw[13], occurs when soil warming melts ground ice, causing land surface collapse[14,15]. Water pooling in collapsed areas leads to formation of taliks (unfrozen thaw bulbs) beneath expanding lakes, accelerating permafrost thaw far faster and deeper than would be predicted from changes in air temperature alone[16–19]. Remote sensing and field observations reveal that localized abrupt thaw features, including thermokarst lakes, thermo-erosional gullies, thaw slumps, and peat-plateau collapse scars, are extensive across northern landscapes with ice-rich permafrost[13]. Despite two decades of observations showing that thermokarst lakes[20,21] and other abrupt thaw features[22] are hotspots of $^{14}$C-depleted permafrost-derived $CH_4$ emissions (Fig. 1), the impact of abrupt thaw on state-of-the-art land-model predicted PCF[6–10] remains unknown.

Here we re-analyzed model output from an earth system model, the Community Land Model version 4.5 (CLM4.5BGC)[7], to identify the carbon-emissions fraction originating from gradual permafrost thaw on land, and to quantify the increase to 21st-century circumpolar permafrost-carbon emissions by including warming-induced abrupt thaw beneath thermokarst lakes. Our abrupt thaw emissions are derived from a vertically- and latitudinally-resolved box model of permafrost carbon processes, including a novel component conceptually describing circum-Arctic thermokarst-lake dynamics and related carbon release[23], which we refer to here as the Abrupt Thaw (AThaw) model (Supplementary Table 1).

CLM4.5BGC and AThaw differ in model complexity, forcing parameters, and resolution (Methods), but both include basic sets of permafrost processes and multiple soil organic carbon pools for projecting permafrost-carbon dynamics[7,23]. CLM4.5BGC focuses on gradual, homogenous thaw affecting only surface permafrost (< 3 m) across the landscape. It also accounts for carbon cycling in the seasonal active layer and stimulation of plant growth by future elevated atmospheric $CO_2$ concentrations and increased nitrogen mineralization with increasing soil organic matter decomposition[7]. In contrast, AThaw simulates the more heterogeneous process of abrupt, deeper (≤15 m) thaw

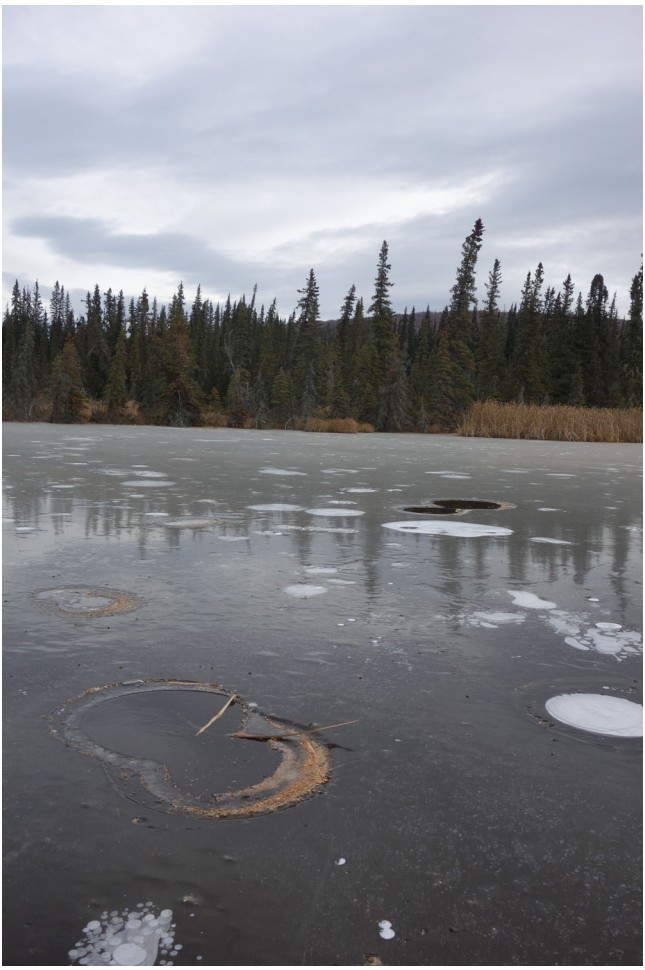

**Fig. 1** Methane bubbling in an interior Alaska abrupt thaw lake. Methane bubbles released by ebullition from thaw bulbs beneath thermokarst lakes are seasonally trapped in winter lake ice forming white bubble patches. High bubbling rates of particularly strong ebullition seeps known as hotspots maintain ice-free holes in winter lake ice. Ebullition hotspots are indicative of abrupt thaw environments, where the rapid (decadal-scale) transformation of terrestrial permafrost to deep thaw bulbs beneath lakes fuels anaerobic decomposition of $^{14}$C-depleted soil organic carbon and the release of $^{14}$C-depleted $CH_4$ in bubbles[20,21]. Diameters of the ice-free hotspot holes shown in this October 17, 2016 photograph are between 0.4 and 0.9 m

beneath newly-formed and expanding thermokarst lakes. AThaw lacks the $CO_2$ fertilization effect since plant uptake of atmospheric carbon is negligible in lakes when permafrost-carbon emissions dominate the first century of lake formation[16,24]; however, processes of lake drainage and carbon fluxes associated with plants colonizing drained lake basins are considered (Supplementary Note 1). Here we summed gradual- and abrupt-thaw carbon fluxes as two independent processes, after avoiding double counting on the ≤ 6% land surface area occupied by abrupt thaw lakes. Fluxes are presented in carbon dioxide equivalents (C-$CO_2$e) to account for the ~28 times larger global warming potential of $CH_4$ at the century time scale ($GWP_{100}$)[12] (Supplementary Table 2). We also calculated the circumpolar permafrost-carbon radiative effect (CPCRE), which is the radiative forcing due to atmospheric perturbations in $CH_4$ and $CO_2$ concentration for these permafrost-soil-carbon flux trajectories. Model results for years 1950–2017 are comparable to observational records from recent decades. Model results also reveal a significant

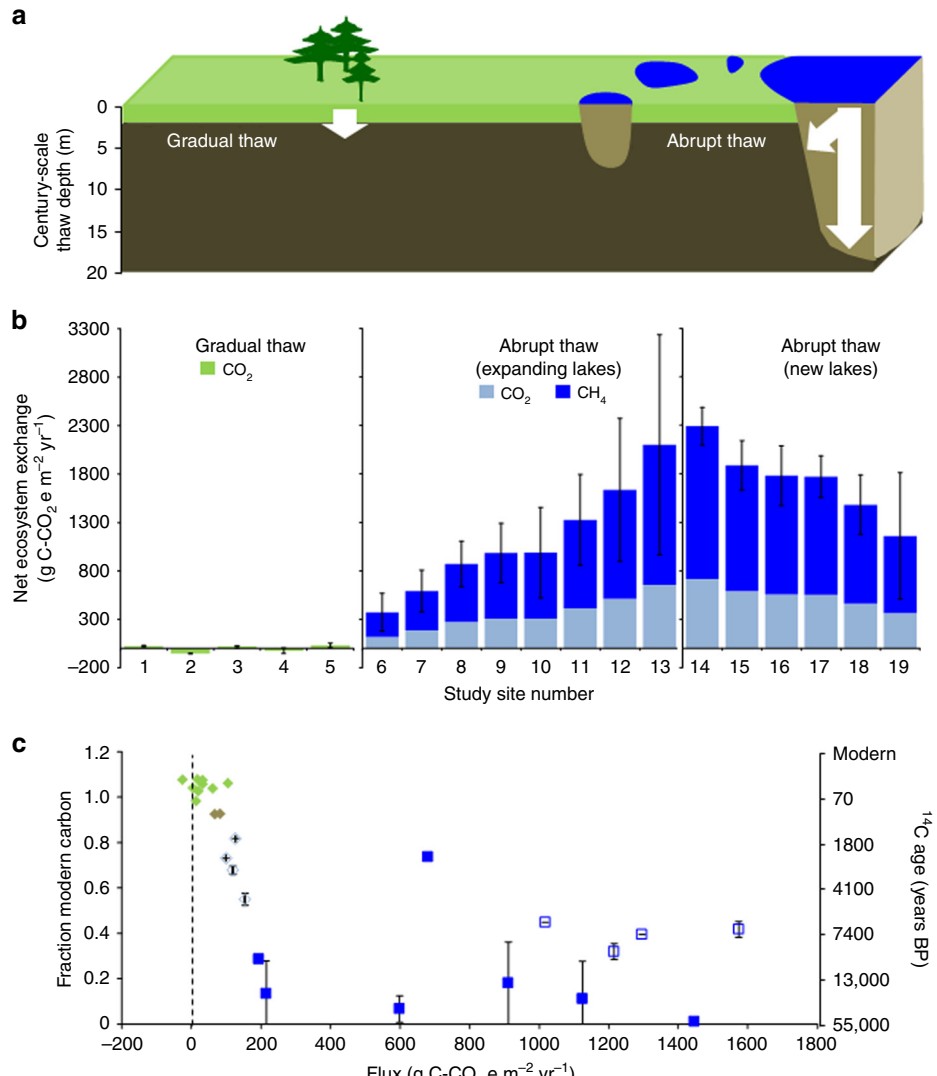

**Fig. 2** Comparison of gradual, top-down thaw in upland permafrost and abrupt thaw beneath lakes. **a** Schematic. **b** Field measurements of net ecosystem exchange (NEE) in Greenland (#1,2)[28], Healy, Alaska (#3–5)[26], western Alaska (#6–9), interior Alaska (#10,11,14–19) and northeast Siberia (#12,13); standard error bars. **c** Radiocarbon-dated $CO_2$ from gradual thaw NEE[26,28] (green), including specifically the old soil carbon component[27] (brown). Radiocarbon-dated $CO_2$ in ebullition bubbles of new abrupt thaw lakes formed since 1949 (blue, open diamonds). Also in **c**, abrupt thaw $CH_4$ emitted as bubbles from expanding lake margins[21] (blue filled squares) and new lakes (blue open squares). In **c**, standard errors of the mean are shown for $n \leq 27$ individually dated ebullition events per lake on 11 lakes (Supplementary Data 2). NEE and radiocarbon ages associated with abrupt thaw were higher than those of gradual thaw (Kolmogorov-Smirnov test, $p < 0.001$)

increase to 21st century circumpolar permafrost-carbon emissions and associated radiative forcing by including warming-induced abrupt thaw beneath thermokarst lakes. These findings demonstrate the need to incorporate abrupt thaw processes in earth system models for more comprehensive projection of PCF this century.

## Results

**Permafrost carbon models and present-day emissions.** While CLM4.5BGC emissions show large interannual variability in the permafrost region's current sink/source status (Supplementary Figs 1 and 2), flux observations and inversion models suggest that the terrestrial permafrost region is an uncertain net $CO_2$ sink (0–0.8 Pg C-$CO_2$e yr$^{-1}$)[25] despite active layer deepening and a few observations of old ($^{14}$C-depleted) soil-carbon respiration observations in Alaska[26,27] and Greenland[28] (Fig. 2b, c). In contrast, abrupt permafrost thaw beneath new and expanding thermokarst lakes is an atmospheric carbon source – Estimates of

present-day emissions by AThaw (7–49 Tg C-$CO_2$e yr$^{-1}$ for years 2011–2017) are similar to observations[21] (19–58 Tg C-$CO_2$e yr$^{-1}$; Methods).

In the context of Holocene-scale thermokarst dynamics, present-day AThaw $CH_4$ emissions (0.7–4.0 Tg yr$^{-1}$, 68% range) represent no significant change from thermokarst-lake emissions over the past 8000 years[29], a pattern that is also consistent with no significant changes in natural arctic $CH_4$ sources during the historical record of atmospheric monitoring[30]. Our 1999–2014 remote-sensing based observation of gross lake-area growth (1.1–1.7%; Fig. 3, Supplementary Fig. 3; Supplementary Table 3) among 73,804 lakes occurring across diverse geographical, climatic, and permafrost regimes in Alaska is at the lower end of the range of gross lake area growth, normalized to the same time scale, that has occurred in various other pan-Arctic regions during the past 60 years[21,31–34] (Methods; Supplementary Table 4). Lacking a longer observational record dissected into multiple time slices and higher-resolution imagery analyses, we

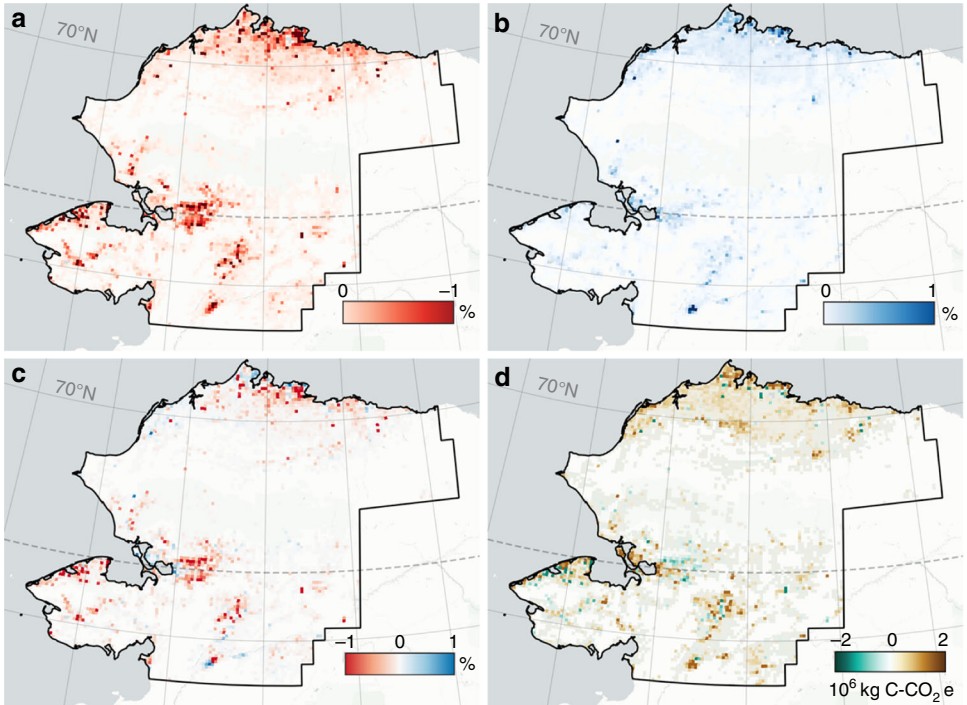

**Fig. 3** Lake area change and carbon flux implications in the permafrost region of northwestern Alaska for 1999–2014 determined with Landsat satellite trend analysis. Gross lake area loss (**a**), gross lake area gain (**b**), net lake area change (**c**), and net change in carbon flux associated with expansion and drainage (**d**) have been aggregated in 7.5 km × 7.5 km grid cells (5625 ha per cell). Background map: CartoDB Positron layer accessed with QuickMapServices QGIS Plugin. Panels **a**–**c** show the absolute percent water fraction change per grid cell. In **d**, net change in carbon flux over the 15-year observation period. Despite a net lake area loss for the region, large permafrost-soil derived carbon emissions associated with gross lake area gain overwhelm smaller flux changes associated with gross lake area loss, leading to a net increase in regional carbon emissions (Supplementary Table 3)

cannot discern this gross lake area increase as different from natural thermokarst-lake processes that occur irrespective of climate warming[31,35,36].

Our observed 1999–2014 gross lake area gain (154 km²) is outweighed by gross lake area loss (i.e., lake drainage, 330 km²) for the same study extent (12,798 km² total lake area; Fig. 3). Nonetheless, this lake change dynamic still contributes and additional 0.9 Tg C-CO₂e of landscape-scale carbon emissions to the atmosphere over the 15-yr study period when field-measured fluxes are applied to increasing and decreasing lake-area changes (Supplementary Table 3, Supplementary Note 2). Despite the net lake area loss, landscape-scale carbon emission to the atmosphere remains positive because conversion of upland terrestrial ecosystems with relatively low carbon fluxes (Supplementary Table 5) to newly formed thermokarst-lake areas with high CH₄ emissions results in a 130- to 430-fold increase in emissions per square meter of land surface change (Supplementary Table 6). The contrasting drainage of lower-emitting older portions of lakes and the establishment of productive, wetland vegetation in drained lake basins leads to smaller changes in carbon fluxes (factor of −0.004 to +0.08). Our net carbon emission estimate is likely conservative because the 30-m resolution Landsat-based analysis did not account for the formation and growth of numerous smaller CH₄-emitting lakes that are only detectable with finer-resolution imagery (Supplementary Fig. 3).

An increase in abrupt thaw lake permafrost carbon emissions requires an acceleration of gross thermokarst lake area growth rates. Such an acceleration was observed in the early Holocene when the frequency of thermokarst-lake basal dates peaked in association with the 1.6 ± 0.8 °C climate warming during the Holocene optimum[29,37]. Widespread acceleration of gross lake area gain during recent decades has yet to be observed among studies of multitemporal satellite imagery[38]; however conclusive

evidence requires quantification of gross lake area growth for large regions using high-resolution imagery in multiple time slices, an analytical combination rarely found in the literature[31,39] (Supplementary Table 4). Analyses of multi-decadal aerial photographs revealed that surface ice-wedge melt, a critical first step in thermokarst-lake formation[40], abruptly increased during the last 30 years in several pan-arctic areas in response to exceptionally warm summers and a long-term upward trend in summer temperature[14,15,41]. Terrestrial Arctic warming of 4–6 °C (RCP4.5) and > 7 °C (RCP8.5) projected to occur this century[10,11] will be unprecedented for the Holocene[37,42] and is anticipated to accelerate gross lake area growth[21,43,44] and the PCF[1,6–9], which could, however, be offset by gross lake area loss resulting from lake drainage[43].

**Abrupt thaw impacts on 21st century emissions.** Our reanalysis of 21st-century CLM and AThaw model results brings to light four critical findings. First, accounting for abrupt permafrost thaw beneath lakes, which peaks mid-century in response to RCP8.5 atmospheric temperature rise (Supplementary Figs 4 and 5), increases late 21st century circumpolar permafrost carbon emissions up to 118% (49–235%, 68% uncertainty range) (Fig. 4b; Supplementary Data 1). Corresponding CPCRE is increased by 130% (62–265%, 68% uncertainty) over that from gradual top-down thaw alone (Fig. 4h), which was the basis of previous PCF assessments[1,6–8]. We acknowledge that the extent to which AThaw emissions affect CPCRE depends on the land-surface models used[6–10], but in all cases would result in a significant increase. End-of-the-century weakening of the abrupt thaw lake contribution to CPCRE occurs because strong warming in RCP8.5 ultimately leads to the loss of high CH₄-emitting lakes through significant landscape-scale drainage[43,45]. Meanwhile,

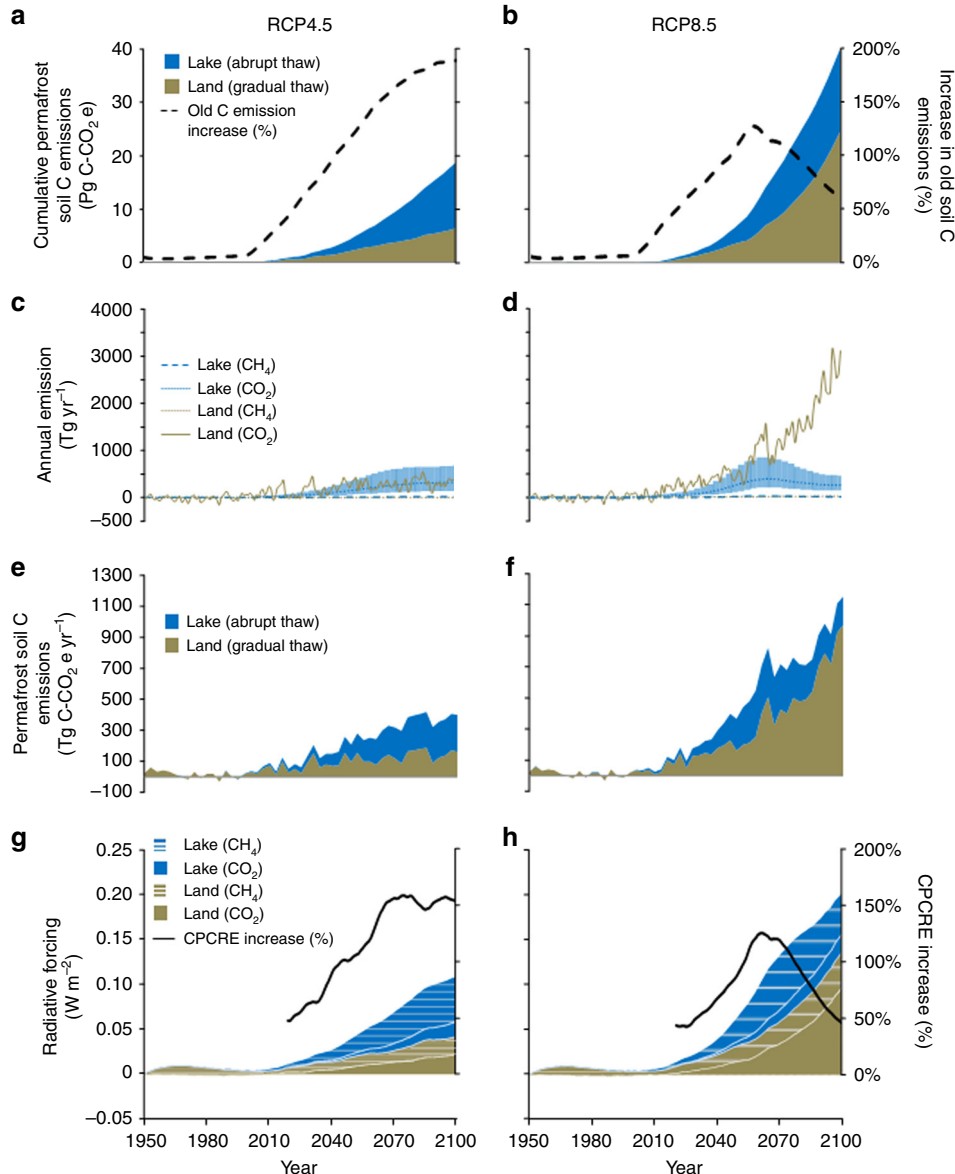

**Fig. 4** Circumpolar permafrost carbon emissions from gradual thaw (land) versus abrupt thaw (lakes) from 1950 to 2100. Permafrost soil carbon emissions, modeled according to representative concentration pathway (RCP) 4.5 and 8.5 scenarios, are distinguished for homogenous gradual thaw from CLM4.5BGC (brown) and heterogeneous abrupt thaw by AThaw thermokarst-lake formation (median values, blue). Cumulative emissions, including percent increase in emissions from old permafrost carbon by abrupt thaw (**a**, **b**). Annual $CH_4$ and $CO_2$ emissions expressed individually as Tg yr$^{-1}$ (**c**, **d**) and collectively as Tg C-$CO_2$e yr$^{-1}$ (**e**, **f**) based on a GWP$_{100}$ of 28 (ref. [12]) and units conversions shown in Supplementary Table 2. Error bars (**c**, **d**) surrounding the median lake emissions are the 68% uncertainty range from a 500 member AThaw model ensemble. Radiative forcing (**g**, **h**) associated with fluxes in **c** and **d**. In **g** and **h**, the increase in circumpolar permafrost-carbon radiative effect (CPCRE) attributed to abrupt thaw lakes is shown only for years 2018–2100, when the AThaw modeled carbon release exceeds one standard deviation in simulated CLM permafrost carbon fluxes during the 1950– 2017 reference period

gradual top-down thaw increases at the end of the century[7] (Fig. 4).

Second, while permafrost-carbon emissions from lakes are a similar magnitude under RCP4.5 and RCP8.5 (Supplementary Fig. 6), the impact of these increased emissions on CPCRE is more pronounced in the moderate forcing scenario (RCP4.5) compared to the strong (RCP8.5) forcing scenario (Fig. 4g, h), intensifying the need for policy makers to take permafrost carbon into account when evaluating climate mitigation scenarios[9]. Including the RCP4.5 median cumulative emissions from new thermokarst lakes [12.3 (5.7–26.7, 68% uncertainty) Pg C-$CO_2$e by 2100] nearly triples what has until now been accepted as the CPCRE from gradual thaw alone (6.4 Pg C-$CO_2$e by 2100),

resulting in a ≤162% increase (76–350%, 68% uncertainty) in circumpolar permafrost-carbon radiative forcing (Fig. 4g, Supplementary Data 1). High AThaw impact on CPCRE under RCP4.5 is caused by differences in the responses of gradual versus abrupt thaw dynamics to moderate climate forcing. In the gradual thaw setting for RCP4.5, atmospheric carbon uptake by plants growing in active layer soils is stimulated more than decomposition of soil organic matter. However, the same degree of warming triggers an acceleration of abrupt thaw via thermokarst-lake formation on up to 4.9% (3.0–6.6%, 68% uncertainty range) of the permafrost-dominated landscape (Supplementary Fig. 4), a pattern consistent with independent thermokarst-lake modeling for northeast Siberia[43].

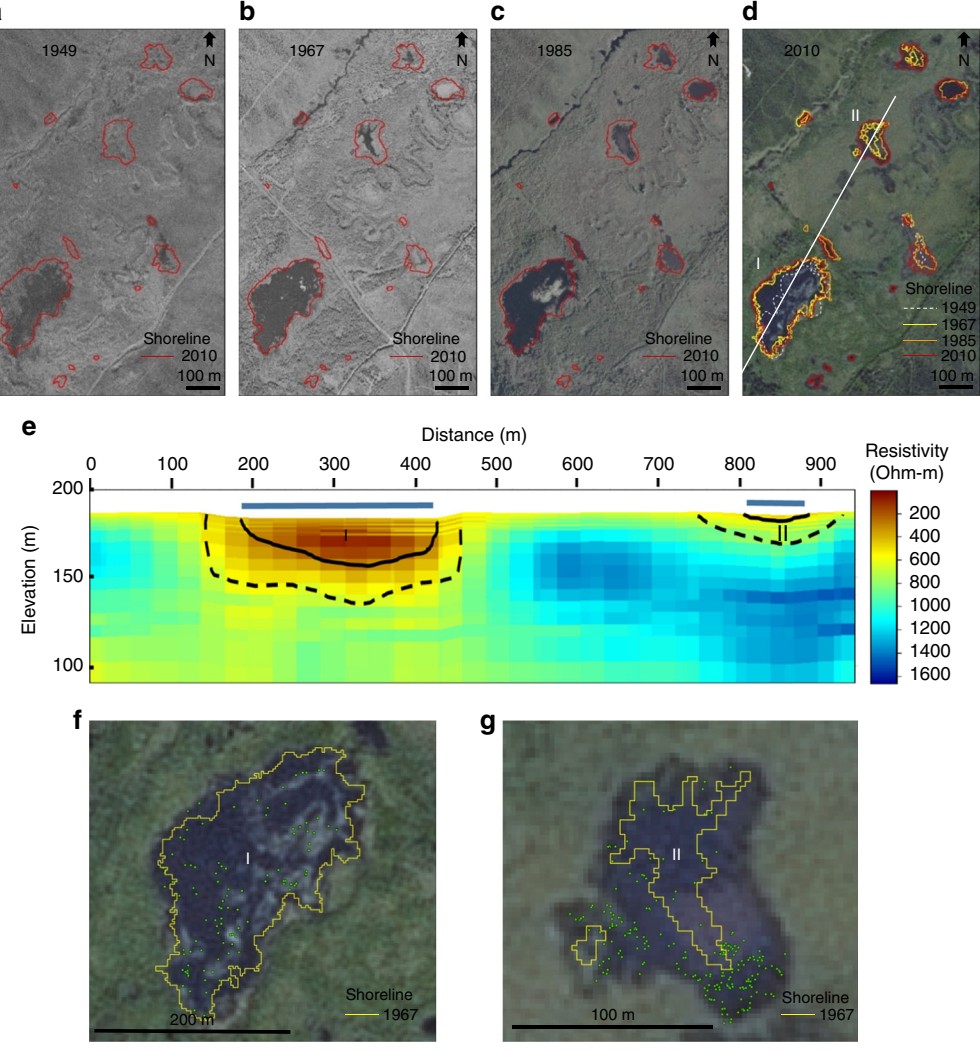

**Fig. 5** Interior Alaska abrupt permafrost thaw. Time series (1949–2010) showing thermokarst-lake development based on historical aerial photograph (**a**–**c**) and 2.5-m SPOT (**d**) shoreline analysis. Dashed lines in **d** delineate dark areas in 1949 images interpreted as shallow, vegetated wetlands that later developed into open-water lakes. In **e**, thaw depth (solid line) and degraded permafrost (dashed line), interpreted from helicopter-borne electromagnetic data-derived 3D resistivity model (Supplementary Methods); extracted cross section is the white line in **d**. Field work shows strong, [14]C-depleted methane ebullition seeps [green dots in (**f**) Lake I, Oct. 2014 and (**g**) Lake II, Oct. 2016] overlapping recent abrupt permafrost thaw areas (e.g., thaw since 1949–1967). Strong seeps are absent in lakes lacking recent abrupt thaw (Supplementary Fig. 8, Supplementary Discussion)

Third, $CH_4$, not $CO_2$, is the dominant driver of the CPCRE, responsible for up to ~70% of circumpolar permafrost-carbon radiative forcing this century (Fig. 4g, h, Supplementary Data 1). On a mass basis AThaw lake $CH_4$ emissions are small relative to $CO_2$ (Fig. 4c, d); however, persistent $CH_4$ emissions and their larger $GWP_{100}$ results in $CH_4$ contributing most of the radiative forcing from abrupt thaw (Supplementary Fig. 6). Methane's contribution to CPCRE may be still higher in CLM simulations where terrestrial soils do not dry following permafrost thaw[46].

Finally, including AThaw lake formation increases cumulative old ([14]C-depleted), permafrost-derived carbon (C-$CO_2$e) emissions up to 127% (RCP8.5) and 190% (RCP4.5) during the late 21st century (Fig. 4a, b). Although climate impact is independent of carbon age, [14]C-flux monitoring is useful for assessing PCF changes and biogeochemical activation of a previously inert, large carbon pool as increasingly old permafrost thaws. While only a fraction of 21st-century gradual-thaw model emissions originates from permafrost thaw itself (10–95% is active-layer-soil carbon mineralization, range depending on land surface model type[7,8], Supplementary Fig. 2), talik expansion beneath lakes accelerates

deep permafrost thaw, mobilizing older permafrost carbon on shorter time scales[16–19].

Combining helicopter-borne electromagnetic (AEM) observations of talik depth in lakes formed since 1949 in Goldstream Valley, interior Alaska with field-based quantification of lake $CH_4$ emissions (Fig. 5, Supplementary Fig. 7), we compared observations of these abrupt permafrost thaw rates and their associated surface fluxes with those of gradual permafrost thaw in the terrestrial uplands reported in the literature. Hotspots of [14]C-depleted $CH_4$ seepage occur in lakes newly formed since 1949 (Figs 1 and 2). In contrast, older lakes that have not expanded lack these hotspots (Supplementary Fig. 8). The vertical thaw depth beneath abruptly-formed thermokarst lakes (8–15 m, Supplementary Fig. 7) was ten to 30-fold greater than the seasonal thaw depth (active layer) in the forested area adjacent to the abrupt thaw study sites (0.5–0.6 m) and in other interior Alaska and Greenland terrestrial sites (0.6–0.8 m)[26,28], where emissions of old permafrost carbon have also been recently observed[26–28]. This ten to 30-fold difference in thaw depth corresponded to a 21–34-fold larger C-$CO_2$e $yr^{-1}$ emission of old

($^{14}$C-depleted) permafrost soil carbon to the atmosphere in the abrupt thaw lake environment compared to gradual thaw in the upland tundra near Healy, Alaska[27]. Comparing abrupt thaw emissions to net surface carbon fluxes (NEE) in the terrestrial uplands near Healy, Alaska[26], and Greenland[28,47], we found a 55- to >2200-fold higher emissions from the abrupt thaw lakes (Fig. 2b) (Kolmogorov Smirnov, $p < 0.001$). The higher emissions relative to thaw depth ratio for abrupt versus gradual thaw may be explained in part by higher mean annual temperatures in talik sediments beneath lakes that remain unfrozen year-round[48] compared to the deepest thawed material at the base of terrestrial active layers, which reach temperatures just above 0 °C for up to a few months per year when these soil layers are not frozen[5]. We acknowledge that talik development will become more wide-spread in the terrestrial environment in the future too, once thaw goes deep enough to separate permafrost from the seasonally freezing surface layer[49]. Nonetheless, the ages of lake-emitted carbon observed among numerous pan-Arctic thermokarst lakes (CH$_4$: 2174–42,900 years BP, median 17,522 years BP, $n = 72$) and among our interior Alaska study lakes newly formed since 1949 (CH$_4$: 6292–10,125 years BP; CO$_2$: 1626–4811 years BP; Fig. 2c, Supplementary Data 2), are up to tens of thousands of years older than old carbon emissions from gradual permafrost thaw on land (CO$_2$: 567–700 years BP, $n = 4$)[27] (Fig. 2c) (Kolmogorov Smirnov, $p < 0.001$) (Fig. 2c). This suggests that including abrupt thaw in PCF scenarios will increase not only the emission magnitude of old carbon, but also its radiocarbon age, improving feasibility for atmospheric $^{14}$C monitoring[50] to detect changes in the permafrost-carbon source.

Whether the warming Arctic will become wetter or drier will impact future PCF strength according to abrupt thaw lake abundance[43] and gradual-thaw CH$_4$/CO$_2$ emission ratios[46]. However, state-of-the-art CMIP5 models consistently predict an increase in precipitation relative to evapotranspiration in the Arctic, especially in summer[51], favoring hydrological conditions for enhanced thermokarst-lake development[43]. Many newly formed lakes will ultimately be subject to drainage[23,43,44] when they intersect topographical drainage gradients by lateral expansion[52], from elevated water levels[53], or when taliks penetrate permafrost, allowing the potential for internal drainage to the groundwater system[54] (Supplementary Figs 4 and 5). While AThaw does not project fluxes in drained lake basins, we consider the implications of lake drainage on landscape-scale fluxes. Present day areal-based carbon fluxes in drained lake basins are one to three orders of magnitude lower than abrupt thaw lake emissions due to refreezing of taliks and colonization of drained basins by plants, whose CO$_2$ uptake offsets emissions[55,56] (Supplementary Table 5). It is conceivable that this difference could be smaller by the end of the century, particularly for RCP8.5, when temperatures are warm enough to prevent refreezing of taliks following lake drainage[49]. Methanotrophy[57] will offset emissions of CH$_4$ produced in drained-lake-basin taliks. However, ecosystem-scale microbial studies show a higher temperature response by methanogenesis than by methanotrophy or by CO$_2$ fluxes attributable to respiration and photosynthesis[58,59]. This indicates that in a warmer world, CH$_4$ emissions and the ratio of CH$_4$ to CO$_2$ emissions from individual ecosystems will increase[59,60]. This also implies that our estimate of AThaw contributions to late-century CPCRE is conservative, particularly for RCP8.5, and would be higher if fluxes in drained lake basins were also taken into account.

**Abrupt thaw implications.** While the cumulative land area subject to abrupt thaw lake formation is less than one tenth of permafrost land areas (Supplementary Fig. 4e, f), our modeling

results (Fig. 4), supported by field work (Figs 1 and 5) and remote sensing (Fig. 3), show that an increase in the volume of newly thawed sub-lake sediments through expansion of existing and formation of new thermokarst lakes is likely to yield dis-proportionately large releases of $^{14}$C-depleted permafrost carbon to the atmosphere this century. The 27 Tg yr$^{-1}$ (15–50 Tg yr$^{-1}$, 68% range) increase in CH$_4$ emissions from newly-formed lakes by mid-century for RCP8.5 (Supplementary Fig. 6), is similar to a recent, independent process-based model estimate by 2100 (27–38 Tg yr$^{-1}$)[44] and is nearly triple the 10 Tg yr$^{-1}$ rise in global human and natural sources of atmospheric CH$_4$ observed from 2003 to 2012[60]. Other non-lake mechanisms of abrupt thaw, such as thermoerosional gullies, thaw slumps, and peat-plateau col-lapse scars, will increase permafrost-carbon emissions further[13].

The moderate climate mitigation strategy (RCP4.5) requires a > 50% reduction in anthropogenic CO$_2$ emissions (i.e., −20 Gt CO$_2$ yr$^{-1}$) by 2100 compared to the current level[61]. Our projected permafrost emissions are comparatively small (1.5–4.2 Gt CO$_2$e yr$^{-1}$ by 2100 for RCP4.5 and 8.5, respectively). However, they are of similar magnitude to the second most important anthropo-genic source after fossil fuels [Land Use Change emissions 3.5 ± 1.8 Gt CO$_2$ yr$^{-1}$], which has been relatively constant during the last 60 years[62], implying that our projected permafrost emissions will provide a headwind in the goal to aggressively mitigate CO$_2$ emissions.

In contrast to shallow, gradual thaw that may rapidly re-form permafrost upon climate cooling, deep, CH$_4$-yielding abrupt thaw is irreversible this century. Once formed, lake taliks continue to deepen even under colder climates[17], mobilizing carbon that was sequestered from the atmosphere over tens of thousands of years. The release of this carbon as CH$_4$ and CO$_2$ is irreversible in the 21st century. This irreversible, abrupt thaw climate feedback is large enough to warrant continued efforts toward integrating mechanisms that speed up deep permafrost-carbon thaw and release into large-scale models used to predict the rate of Earth's climate change.

## Methods

**Summary of the modeling approach.** Permafrost-region 21st-century soil carbon emissions are compared between two model types using IPCC RCP4.5 and RCP8.5. Both models represent basic sets of permafrost processes and have multiple soil organic carbon pools, but the models differ in their complexity of how individual processes are described. While CLM4.5BGC[7] simulates the full physical interac-tions between the atmosphere and the soil, including vegetation uptake of CO$_2$, AThaw parameterizes soil thaw rates, depending on ground thermal properties, mean annual ground temperatures, active layer depth, and magnitude of the regional warming anomaly which drives permafrost degradation[23]. In its current model design, CLM4.5BGC simulates gradual thaw in terrestrial uplands. We utilized CLM4.5BGC emission data for years 1950–2100 partitioned according to non-permafrost carbon, originating from present-day active layer horizons, and permafrost carbon, which becomes thawed from the top down as active layer gradually deepens. In contrast, the more simplistic, but therefore more flexible model design in AThaw also allows for abrupt thaw under thermokarst lakes by tuning model parameters to simulated talik growth rates of a physically-based thermokarst-lake model[16]. Newly-formed thermokarst lakes and laterally-expanding lake margins are a large net source of atmospheric CH$_4$ and CO$_2$, in contrast to mature thermokarst-lake stages, where fluxes are lower (Supplementary Tables 5 and 6). AThaw simulates only the carbon emissions from newly thawed sub-lake sediments comprising contributions from expansion of thaw lakes present since 1850, and from new lake initiation in response to warming after the year 1850. AThaw does not simulate fluxes for older lake areas already present on the landscape at year 1850. Further, permafrost degradation in CLM4.5BGC is driven by spatially resolved climatic forcing fields, while AThaw focuses on large-scale latitudinal climatic gradients. In addition to differences in spatial resolution, CLM4.5BGC is more complete, simulating a large set of climate variables which all determine surface vegetation and soil conditions, while AThaw considers surface air temperatures only as the key driver for the net balance of lake expansion versus drainage.

**AThaw model.** AThaw is a conceptual model which projects carbon release from abrupt thaw by accounting for the full chain of processes from formation of new

thermokarst lakes under global warming and talik deepening in sub-lake sediments to eventual carbon release to the atmosphere following anaerobic microbial degradation of organic matter and $CH_4$ oxidation. AThaw also accounts for lake drainage; although, carbon fluxes in drained lake basins are not modeled[24,63–65]. AThaw is incorporated into a multi-box permafrost-carbon release model which allocates soil organic matter into latitudinally and vertically gridded boxes of differing conditions regarding soil physics, carbon quantity and quality, and biogeochemistry[23].

Briefly, AThaw model resolution consists of (a) 20 latitudinal bands, ranging from 45°N to 85°N with a 2° gridding, and of (b) 27 vertical soil layers corresponding to layer thicknesses of 25 cm for the upper 4 m, and of 1 m thickness for the depth range of 4–15 m. AThaw assumes typical soil organic matter residence times in permafrost soils, which are determined by partitioning into passive, slow, and fast cycling carbon pools with decomposition timescale parameters based on incubation experiments. $Q_{10}$ temperature sensitivity is accounted for as well as $CH_4$ oxidation [for details see Supplementary Table 1 and Schneider von Deimling et al.[23]].

AThaw assumes that increasing Arctic temperatures will drive expansion of existing lakes and new thermokarst-lake formation by melting of near-surface ground ice and subsequent ground subsidence. This assumption is in line with Community Land Model (CLM4.5) results from Lee et al.[66], whereby surface excess ice in permafrost soils of many regions will largely melt by 2100 when subject to intense warming. AThaw assumes that comparatively small 19th and 20th-century warming rates have initiated some formation of new but rather shallow thermokarst lakes with likely winter-refreeze of lake bottom sediments. Start of abrupt thaw (i.e., formation of sub-lake taliks) is only assumed to occur for stronger warming starting in the 21st century and beyond. The evolution of newly-formed thermokarst lakes is parameterized in AThaw by an optimum function which non-linearly scales the latitudinal thermokarst lake area fraction by the surface air temperature anomaly[23] (Supplementary Figs 4 and 5). The lake formation scheme models an increase of newly-formed thermokarst-lake areas with temperature until a maximum extent [$F^{TKLmax}$ (~8 to ~40% increase, depending on soil type)], is reached under a temperature optimum $dT'^{TKLmax}$. The temperature optimum corresponds to high-latitude surface air temperatures 4–6 °C above pre-industrial; warming above this optimum shifts 21st century lake dynamics toward net drainage. Such responses of thermokarst lake formation to temperature increases are not unprecedented, given that the early Holocene Arctic temperature increase of 1.6 °C[37] resulted in a 570% increase in thermokarst-lake formation rates[24]. AThaw-projected increases in lake area are lower for 2010–2100 compared to the early-Holocene observational record, because permafrost soils are now more protected from warming and thawing by thick organic soil surfaces[67] and less likely to form large lakes compared to the less dissected early Holocene periglacial landscape[52].

**Temperature-driven thermokarst lake dynamics.** Rather than simulating individual lake life cycles of formation, expansion, drainage, and re-initiation of later generation lakes (e.g., ref. [43]), AThaw quantifies the net effect of new lake formation and drainage in modelled lake areas ($F^{TKL}$). We capture a wide range of uncertainty in net lake formation and drainage trajectories by varying two key model parameters: The maximum net lake area, $F^{TKLmax}$, and the optimum high latitude surface temperature increase, $dT'^{TKLmax}$ (Supplementary Fig. 5). Model parameters for the maximum lake area extent ($F^{TKLmax}$) were chosen individually for the four AThaw soil classes to capture expected differences in the potential for future lake development (see Methane and $CO_2$ in newly-formed thermokarst lakes). However, the parameter that most strongly controls the dynamics of thermokarst lake formation and drainage is the temperature optimum, $dT'^{TKLmax}$, the temperature at which the maximum lake area occurs. We prescribed a mean estimate of 5 °C for $dT'^{TKLmax}$ (i.e., high latitude surface air warming above pre-industrial) and consider a full range of 4–6 °C in our model ensemble. This parameter choice is based on paleoenvironmental evidence of Early Holocene warming by a few degrees Celsius in Northern Hemisphere land areas[37,68,69] which resulted in rapid and intensive thermokarst activity[24,29,70].

This range of $dT'^{TKLmax}$ values was also chosen to capture the sensitivity of future surface ice-wedge melt to climate warming, since ice-wedge melt is a critical step in thermokarst lake formation and drainage. Increases in permafrost temperature, which typically mirror increases in air temperature, have been observed in many Arctic regions, with warming of up to 3 °C since the 1970's in some of the coldest permafrost regions[3,38,71]. Permafrost warming is often accompanied by active layer deepening and increased ground-ice melt. Widespread surficial degradation of ice wedges and a significant increase in areas of water-filled polygonal troughs has been linked to climatic warming during the past few decades in northern Alaska, the Canadian Arctic Archipelago, and Siberia[14,15,41]. Additional permafrost warming on the order of 2–3 °C anticipated by 2050 for RCP4.5 and RCP8.5[49], is expected to intensify permafrost and surface ice wedge degradation[66], enhancing thermokarst. For RCP4.5, permafrost warming slows beyond 2050[49], supporting AThaw ensembles of lower net thermokarst lake formation during the latter part of the century (Supplementary Figs 4 and 5). In contrast, extreme warming after 2050 for RCP8.5 is expected to heavily degrade permafrost, such that near-surface permafrost disappears entirely from many arctic

regions[49]. These conditions lend support to AThaw parameterization, which leads to net lake drainage in the later part of the century for RCP8.5.

In AThaw, warming above the temperature optimum in the 21st century is assumed to lead to a reduction in AThaw lake area due to increasing lake drainage[23,43,45], until a prescribed minimum fraction $F^{TKLmin}$ remains. The minimum fraction of the landscape still covered by newly formed lakes, decreases from north to south[23]. In northern, continuous permafrost regions, lakes in AThaw are prescribed to drain laterally as melting of the ice-wedge network in the surface surrounding lakes can create drainage pathways[52]. Other mechanisms of lake drainage include bank overtopping, headward gully erosion towards a lake; and tapping of lakes by streams, rivers, or other water bodies[36]. Higher drainage potential in southern, discontinuous permafrost regions also encompasses internal drainage of lake water through open taliks that penetrate thin permafrost in groundwater recharge settings[54].

While the 4–6 °C $dT'^{TKL}$ range used in our modeling was prescribed based on paleoenvironmental evidence, historical observations, and modeling of future permafrost dynamics, we can consider the implications of using smaller or larger $dT'^{TKLmax}$ values. Smaller $dT'^{TKLmax}$ values would imply that the maximum area of thermokarst lake coverage would occur earlier in the century. This would result in a relatively larger AThaw increase to the PCF earlier in the 21st century and a lower AThaw contribution to PCF toward the end of the century due to more widespread drainage. If $dT'^{TKLmax}$ aligned more closely with the range of simulated high latitude warming inferred from higher concentration pathways (e.g., $dT'^{TKLmax} > 6$ °C), we would expect maximum thermokarst lake coverage (i.e., $F^{TKLmax}$) to occur later in the century, less drainage of AThaw lakes during the 21st century, and a larger relative contribution of AThaw to end-of-the century PCF under stronger future warming.

It is interesting to note that that while thermokarst-lake initiation and drainage rates are linked to climate, thermokarst-lake growth—the long process between initiation and drainage that results in most of the carbon release—has dynamics (e.g., talik growth, shore expansion) that once started become rather decoupled from climate due to strong linkage with local factors such as ground ice content and ice-layer thickness[72–75]. Hence, thermokarst lakes are found across the entire range of Arctic climatic zones and permafrost temperatures as long as there is sufficient ground ice[36,76,77]. A thermokarst lake on the New Siberian Islands has the same potential to release carbon as a thermokarst lake in Central Yakutia; the differences are largely not determined by climate (or RCP conditions) but by local conditions such as permafrost soil carbon and ground ice contents. Hence, either RCP scenario will result in more lakes (earlier or later) and both scenarios will have a similar emission magnitude linked to maximum lake areas, but relative to anthropogenic emissions, the thermokarst lake emissions from RCP4.5 will be more relevant.

Factors other than temperature that are not included in AThaw, but which can also affect thermokarst lake dynamics include natural and anthropogenic surface disturbance, precipitation changes, and local topography[43]. However, since CMIP5 models consistently predict a moistening trend in the Arctic (i.e., an increase in precipitation relative to evapotranspiration)[51], an explicit accounting for predicted 21st-century precipitation and evaporation trends would reinforce rather than weaken the AThaw lake dynamics driven by temperature changes alone.

**Methane and $CO_2$ in newly-formed thermokarst lakes.** AThaw simulates pan-Arctic $CH_4$ ($CH_{4TKL}$) and $CO_2$ ($CO_{2TKL}$) release from thermokarst lakes proportional to the amount of newly-thawed carbon that becomes vulnerable to microbial decomposition. The volume of this newly-thawed carbon expands vertically by talik growth in sub-lake sediments and horizontally by increases in the extent to which thermokarst lakes cover the landscape. Here we discuss how these two processes, vertical and horizontal growth, are captured in AThaw and how they compare to other modeling studies and observational evidence.

First, AThaw simulates vertical talik growth rates beneath lakes as a function of atmospheric temperature anomalies, which in turn determine lake bottom temperatures and ultimately drive sub-lake sediment warming. Thaw rates are assumed to depend on soil ice contents, mean ground temperature, depth of the thaw front, and are tuned to reproduce talik deepening simulated by Kessler et al.[16]. For instance, AThaw simulates typical talik depth beneath lakes of 11 m (7.8–13.4 m, 68% uncertainty range; 20 m max) in warm permafrost environments (i.e., mean annual soil temperatures close to 0 °C) with mineral soils by the year 2050 under RCP8.5. Our helicopter-borne electromagnetic measurements of sites with comparable climatic and soil conditions (i.e., relatively warm permafrost temperatures between −3 and −1°C) (Supplementary Information) have inferred lake talik growth of 8–15 m in < 50 years (Supplementary Fig. 7). It should be noted that advective heat transport by groundwater can accelerate vertical thaw; maximum reported thaw rates are up to a meter per year[78,79]. When accounting for the effect of ground subsidence in a modeling context, Westermann et al.[79] suggest that several meters of ground ice can be removed in less than a decade. The authors simulated a talik growth of 15 m by 2100 under RCP8.5 for a site characterized by yedoma ice complex in cold, continuous permafrost. In contrast, AThaw simulates 6 m (4–8 m, 68% range) talik depth under comparable climatic conditions by end of the century. Therefore, given modeling and observational evidence, we consider that the thaw rates in sub-lake sediments simulated by AThaw are relatively conservative.

A second key AThaw model assumption concerns simulated thermokarst-lake expansion in a warmer climate. Central to this aspect is the question to which extent newly-formed thermokarst lakes will cover degrading permafrost landscapes in a warmer climate. AThaw assumes that future thermokarst-lake formation will strongly depend on soil conditions (e.g., ice content) and landscape morphology. Therefore, the model assumes different maximum lake formation extents for four different ice-rich soil type distributions in the permafrost region of the Arctic: mineral soil (Orthels and Turbels), organic soils (Histels), undisturbed yedoma, and refrozen thermokarst deposits in the yedoma region (Supplementary Table 1, Supplementary Figs 4 and 5). Orthels, Turbels and Histels follow the Northern Circumpolar Soil Carbon Database (NCSCD)[80], while undisturbed yedoma and refrozen thermokarst deposits in the Pleistocene-aged ice-rich yedoma soil region are distinguished according to Strauss et al.[81] and Walter Anthony et al.[24].

For mineral soils, AThaw assumes that newly-formed lakes can degrade a maximum fraction $F^{TKLmax}$ of 7% (4–9%, 68% range; Supplementary Fig. 4) of the landscape. Given the large-scale dominance of mineral soils in the permafrost region soil carbon inventory (540 Pg C, NCSCD)[80], this soil class contributes significantly to cumulative pan-arctic $CH_{4TKL}$ (at year 2100, 46% for RCP8.5, 28% for RCP4.5). Organic soils (120 Pg C, NCSCD)[80] are typically richer in ground ice than mineral soils and therefore AThaw assumes a factor two larger potential for the formation of new thermokarst lakes [$F^{TKLmax}$ of 14% (10–17%, 68% range; Supplementary Fig. 4)]. This soil type contributes 17% for RCP8.5 and 14% for RCP4.5 to cumulative pan-arctic $CH_{4TKL}$. AThaw explicitly accounts for ice- and organic-rich soils in yedoma regions, separated into undisturbed yedoma landscapes (80 Pg C)[81] and drained lake basins in the yedoma region that subsequently formed permafrost following lake drainage (refrozen thermokarst; 240 Pg C)[24,81]. Given the high ice contents of these yedoma-region soils[81–83], both soil classes are assigned a high potential for the formation of new thermokarst lakes once rising temperatures have resulted in ground subsidence. Given the high volumetric ice contents of refrozen drained thermokarst lake basins (> 50%)[81,83] in conjunction with its basin-type geomorphology, which favors water ponding[16], AThaw assumes a maximum $F^{TKLmax}$ of 21% (11–27%, 68% range; Supplementary Fig. 4). Refrozen thermokarst basins in the yedoma region cover only about 4% of the permafrost domain of the Arctic, but they contribute 28% under RCP8.5 (38% under RCP4.5) to cumulative pan-arctic $CH_{4TKL}$ by 2100. The highest potential for new-lake formation is assumed for unaltered ice-rich yedoma soils, where volumetric segregated ice content is typically 47–53% and high wedge-ice volumes (~40%) further increase permafrost ice concentration[81]. Here AThaw assumes a $F^{TKLmax}$ of 33% (16–42%, 68% range; Supplementary Fig. 4). Considering that thermokarst activity in ice-rich regions had degraded about 80% of the landscape in specific regions during the Holocene[24,81,84], we consider a factor two reduced lake-forming potential plausible, especially under the assumption of strong future warming. While undisturbed yedoma landscapes constitute our carbon pool with the smallest areal extent (covering 0.41 million km$^2$, 2% of the permafrost domain)[81], they contribute 9% under RCP8.5 (20% under RCP4.5) to cumulated pan-arctic $CH_{4TKL}$ by 2100, revealing the importance of carbon release from deep deposits beneath abruptly-formed yedoma thermokarst lakes.

We derived estimates of present-day abrupt-thaw emissions from AThaw and independently from upscaling observations. The estimated emission range from AThaw [20 (7–49) Tg C-$CO_2$e yr$^{-1}$] represents the median and 68% uncertainty range for years 2011–2017 (Supplementary Data 1). The observation-based estimate (19–58 Tg C-$CO_2$e yr$^{-1}$) was derived by upscaling observed lake carbon fluxes from abrupt-thaw features (Fig. 2b) to the extent of observed gross lake expansion areas (5–15%) among pan-arctic regions during the past 60 years[21]. This range of gross lake expansion is comparable to our observation of gross lake-area increase in northern and western Alaska (i.e., 1999–2014, 1.1–1.7% gross lake area increase upscaled to 5–7% assuming similar expansion/formation rates over the past 60 years; Supplementary Table 3, Fig. 3). We acknowledge that present-day emissions associated with abrupt thaw may be conservative since gross lake-area increases in some regions are higher (e.g., > 50% in Quebec[85] and > 23% in West Siberia[86]), potentially due to hydrological changes in addition to permafrost thaw.

The formation/expansion of new thermokarst-lake areas and subsequent sub-lake talik growth is not the only mechanism of abrupt permafrost degradation making newly thawed permafrost carbon available for microbial decomposition. Further contribution comes from talik growth of present-day lakes which have not yet formed a deep talik and still store large amounts of labile carbon in sub-lake sediments. For instance, on the Alaska North Slope, Arp et al.[17] demonstrated that a rapid decrease in lake-ice thickness and duration has already led to many shallow lakes transitioning from bed-ice fast lakes underlain by permafrost to floating ice lakes that have started to develop taliks. This talik formation takes place about 70 years before talik formation is projected for the adjacent terrestrial environment by top-down permafrost models in this cold continuous permafrost zone, a feedback process also not accounted for in AThaw or other models of permafrost degradation. These and other non-lake modes of abrupt thaw[13], including coastal and river erosion, thermoerosional gully formation, thaw slumps, collapse of permafrost peat plateaus, and talik formation in upland terrestrial environments, which may occur sooner than predicted by large-scale models when finer resolution soil and vegetation properties are taken into account[49], are not explicitly accounted for in the AThaw model description; hence, we consider our assumed $F^{TKLmax}$ values a conservative estimate of the abrupt thaw extent in the 21st century.

**Uncertainty**. Uncertainty in AThaw modeled carbon fluxes is based on independent sampling of a set of 18 key model parameters which are subject to either observational or to model description uncertainty[23] (Supplementary Table 1). For each warming scenario (RCP4.5 and RCP8.5), 500 ensemble runs were performed by applying a statistical Monte Carlo sampling and by assuming uniformity and independence in the distributions of model parameters and initial values. AThaw results are presented as the median and 68% uncertainty range. We acknowledge that our ability to accurately quantify uncertainties is limited given the use of this single model and its assumptions in a highly complex and large system.

**Double counting**. To avoid double counting CLM4.5BGC emissions from land areas that become thermokarst lakes in AThaw, and to account only for the increase in emissions and CPCRE caused by abrupt thaw, we have subtracted from the AThaw emissions and CPCRE, in our total permafrost landscape calculations, the quantity of carbon emissions and associated radiative forcing already assumed to be emitted from those land surfaces from CLM4.5BGC. We used the AThaw thermokarst-lake area fraction at each time step for each of the four soil classes (Supplementary Fig. 4) and weighted those fractions by the areal extent of the soil classes according to Hugelius et al.[80] and Strauss et al.[81] based on the implicit assumption of homogeneous soil carbon distribution within each of the soil classes. Our calculations consider that this fraction of the land surface, subject to gradual thaw in CLM4.5BGC, undergoes abrupt thaw instead.

**Additional methods**. We calculated the radiative forcing due to atmospheric perturbations in $CH_4$ and $CO_2$ concentration for CLM and AThaw permafrost-soil-carbon flux trajectories following Frolking & Roulet[87] (Supplementary Methods). Methodology underlying our remote-sensing based quantification of abrupt thaw in Alaska and field-based estimates of carbon emissions from abruptly-formed thermokarst areas of lakes in Alaska and Siberia are also provided in Supplementary Methods. Field and lab measurements include bubble-trap observations of ebullition fluxes, aerial and ground-based ebullition seep-mapping, and quantification of $CH_4$ and $CO_2$ concentrations and radiocarbon dating.

**Statistics**. To test differences between gradual thaw and abrupt thaw net ecosystem exchange (Fig. 2b) we used the two-sided Kolmogorov-Smirnov test. We used this test also to compare radiocarbon ages of permafrost soil carbon respiration in gradual thaw versus abrupt thaw environments (Fig. 2c) Radiocarbon statistical analysis was performed on percent modern carbon data. All statistical analyses were performed in R[88].

**Data availability**. CLM data are publicly available at the National Energy Research Scientific Computing Center archive (https://www.portal.nersc.gov/archive/home/c/cdkoven/www/clm45_permafrostsims/permafrostRCN_modeldata). AThaw data and calculated radiative forcing for CLM and AThaw fluxes presented in this study are available within the article's supplementary information file (Supplementary Data 1). Radiocarbon data are provided in Supplementary Data 2. All other data that support the findings of this study are available from the corresponding author (K.M.W.A.) upon request.

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

## Acknowledgements

Charles Koven provided data from the Community Land Model, version 4.5 (CLM4.5BGC). Geographic Information Network of Alaska (GINA) provided historical airborne images and orthomosaicked SPOT images. Landsat data was provided through USGS EROS Data Center. Jessie Cherry helped with the aerial image acquisition of ice-covered lakes. Lisa Wirth rectified historical airborne images. Kerstin Schlobies mosaicked recent airborne ice-covered lake images. Burke Minsley created EM1DFM resistivity models. Melanie Engram, Allen Bondurant, and Joanne Heslop helped with field work. C. Koven, David Lawrence, and Dave McGuire provided constructive comments on the manuscript. This work was supported by NSF ARCSS 1500931, NSF ARC-1304823, NASA ABoVE NNX15AU49A, and NASA ABoVE 1572960. G.G., I.N. and T.s.v.D. were supported by ERC #338335, HGF ERC-0013, and ESA GlobPermafrost. T.s.v.D. was further supported by BMBF #03G0836C and BMBF #01LN1709A. S.F. was supported by DOE grants DE-SC0010580 and DE-SC0016440.

## Author contributions
K.M.W.A. conceived of the study and wrote the paper. T.S.v.D. performed AThaw modeling. S.F. created the radiative forcing model; P.A. applied the model and performed statistics. I.N., P.L. and G.G. conducted remote-sensing analyses. R.D. and A.E. are responsible for AEM data and interpretation. K.M.W.A and R.D. conducted field work. B.J. assisted with contextualizing results related to lake drainage versus expansion. All authors commented on the analysis, interpretation and presentation of the data, and were involved in the writing.

## Additional information

**Competing interests:** The authors declare no competing interests.

