## [Peer Review File · Nature Communications]

Reviewers' comments:

Reviewer #1 (Remarks to the Author):

The authors set out to provide an analysis of the permafrost carbon feedback to the climate system under two future scenarios (RCP4.5 and RCP8.5), using the AThaw model to represent the influence of temperature on CO₂ and CH₄ fluxes resulting from changes in the size and condition of thermokarst lakes. These results are contrasted with the output of the CLM4.5BGC land model, which represents the influence of warming on permafrost through changes in active layer thickness but which excludes any biogeochemical processes associated with thermokarst lake dynamics.

The results shown here suggest that consideration of thermokarst lake dynamics leads to significantly higher net carbon emissions and associated radiative forcing. Significant differences in this effect are seen between the RCP4.5 and RCP8.5 scenarios, with a higher (percentage) influence of thermokarst lake dynamics on radiative forcing for RCP4.5 than for RCP8.5. CH₄ contributions to the thermokarst lake radiative impacts are estimated to be stronger than CO₂ contributions. Thermokarst lake dynamics also generate net carbon fluxes with older 14C dates, which may turn out to be a useful observational metric to track how permafrost and thermokarst dynamics evolve in the coming decades.

I have one major concern with the study, and several smaller comments. The major concern is that there appears to be a very critical sensitivity in these results to the assumptions about temperature thresholds and rates for the process of thermokarst lake drainage under conditions of deep talik formation. This sensitivity is acknowledged by the authors in the main text (lines 124-127) and in the methods (lines 422-425). The peak behavior under RCP8.5, and the fact that RCP4.5 sees a stronger percentage influence of thermokarst lake dynamics are both tied directly to the relative temperature thresholds and parameterized rates of change for thermokarst lake expansion vs. thermokarst lake drainage. From the literature cited, it is not obvious that there is a dominance of one process over the other for historical or present-day conditions, or that, under the currently experienced warming, expansion is overtaking drainage as a signal that can be attributed to anthropogenic climate change.

Based on the mentions in the text, I expected to see this as a prominent feature of the parameter uncertainty quantification exercise, but I am not able to identify the parameters controlling thermokarst lake drainage in Table S1. I think it is imperative to carry out this sensitivity analysis, with a reasonably large range for the parameter uncertainties, to show what impact these assumptions have on the four main conclusions of the study. The uncertainty quantification exercise is commendable, but will be more relevant to the most interesting results if expanded in this way. Also, there should be more discussion of how the default parameters for this part of the model are derived.

Other comments:

Lines 55-65: would the mechanisms described here also apply to water accumulation in polygonal tundra, at spatial scales smaller than what would normally be considered a lake?

Fig S3: Are there no examples in this region of lake contraction, due to lake drainage or other processes? I was surprised that the scale bar shows only lake expansion fraction.

Lines 190-192: The statement that, once formed, abrupt thaw features continue to develop even under shifts to a cooler climate, seems to leave out consideration of the potential for lake drainage.

Reviewer #2 (Remarks to the Author):

This is an impressive study focussed on predicting future greenhouse gas release resulting from abrupt thawing of permafrost beneath thermokarst lakes. To my knowledge, a number of recent studies have pointed towards thermokarst areas being particularly significant for permafrost carbon feedback. But this is perhaps the first study to attempt to quantify the emissions, which is largely a modelling exercise using a combination of field measurements, remote sensing and radiocarbon analysis. I think there will be a lot of interest in this study, and it will strongly influence research into permafrost carbon feedback.

I have been asked to comment on the radiocarbon analyses in this study which represent a relatively minor part of the study. I don't comment on the modelling aspects which are largely outside my field anyway.

The radiocarbon analyses undertaken for this study were all performed on samples of methane-rich gas (radiocarbon measurements of carbon dioxide are reported from a published study (Fig. 1c)). The methods used to measure the samples are reasonably routine and the description in the Supplementary Information (SI) indicates that care was taken during the collection, preparation and radiocarbon measurement of these samples. I therefore have high confidence in the measurements.

I think it is a pity that radiocarbon measurements of carbon dioxide were not undertaken as part of this study. I appreciate that the CO₂ concentration in the samples was much lower than methane, and less important in terms of radiative forcing. But I think some analyses of CO₂ would have improved the study. As it is, Fig 1c seems odd, because in this ¹⁴C comparison between gradual-thaw sites and abrupt-thaw sites the former are for CO₂ and the latter are for CH₄. Therefore it's not a like-for-like comparison because the terrestrial CO₂ will have a much greater influence of modern carbon from plants (e.g. Schurr et al. measured ecosystem respiration). If CH₄ had been dated at the terrestrial sites it is very possible that it would have been even older than the terrestrial CO₂.

Some of the radiocarbon concentrations appear to be very close to the limit of detection for ¹⁴C, which to me raises the question of whether the methane is derived from the breakdown of organic material in the permafrost, or whether it could be thermogenic. If there is likely to be organic matter that is >40,000 years old then the latter explanation is not required. However, I think a contribution from a thermogenic source would have implications for the interpretation of the methane fluxes. For example, the observation is reported that methane emissions in older thaw lakes are lower than in younger thaw lakes because the labile fraction of older lakes has been mineralised. Could an alternative interpretation be that much of the methane is thermogenic, built up over time and trapped in the permafrost, and that thawing just allows its release (explaining the extremely low ¹⁴C concentrations)? How would this effect the model predictions?

Figure 1 c. There are 11 data-points for methane radiocarbon concentrations plotted. In the SI (p 8) it says that "72 ebullition events from 11 thermokarst lakes ..were dated". Were individual ¹⁴C analyses undertaken on each ebullition event and the results combined, or were only 11 samples analysed by combining the ebullition events for each lake? Some clarification of this would be helpful.

I think it would be valuable to other researchers if the individual radiocarbon results were reported in the SI, along with any ancillary measurements such as %CH₄ and delta ¹³C. If delta ¹³C was measured then the results should inform on whether there was any thermogenic methane or not.

Figure 1c. Left-hand-side y-axis title should I believe be Fraction modern carbon (rather than Percent modern carbon) – otherwise the ages would all be old and the right-hand-side x-axis wouldn't be correct.

Reviewer #3 (Remarks to the Author):

With only general knowledge of carbon emissions related to permafrost degradation, I learned quite a bit reading this interesting, concise (at least the main part), well written, and important paper. The authors make a strong argument for the role abrupt-thaw permafrost beneath predicted new and expanding thermokarst lakes will play in increasing greenhouse gas concentrations as the climate warms, and suggest that these contributions are large enough to be included in better models of greenhouse gas trajectories through the 21st century. Although principally a modeling effort (and I would quibble about the title lacking any mention that this is a synthetic study), the authors include large amounts of field data, some relevant remote sensing, and even some airborne geophysics to the mix. As a result, I have no major complaints about the paper and feel it would be a worthy addition to the permafrost and climate-change literature. I guess the greatest amount of heartburn from the paper is the indiscriminate mixing of reality and model results; we all recognize models have their limitations and the future is unlikely to unfold as the authors might predict. Some effort to more clearly emphasize that, starting with the title, might be appropriate. Also, it seems that most of the model-based projections differ greatly from recent and historical experience, and those effects begin rather soon. Why have we not seen strong beginnings of these effects in recent decades, or if we have, emphasizing those would help bridge the belief gap between experience and model projections.

Being a non-expert on the principal focus of the paper, I can't find much fault with the approach, results, or interpretations. There are a few places where more clarity would be helpful to a general scientific audience, and there are embarrassingly few other issues I found. The ones I saw are listed below by line number or page number for the supplemental material.

Abstract: Well written and concise.

line 30: Given that the RCPs are concentration pathways, would it be better to say moderate concentration rather than moderate warming? We're sure about the concentration path, but I think the warming magnitude is more uncertain.

lines 103-108: Here's one place where the Holocene record to 8 ka and the historic record show no apparent acceleration in thermokarst dynamics, yet the models predict those will begin "real soon now." 1.2 to 1.8 percent gross lake area growth since 1999 doesn't seem to be a strong argument for acceleration either, although the authors follow that with much larger gross growth rates in other Arctic areas. Please also clarify what you mean by gross lake area growth; the net growth shown in Table S2 shows lake-area reductions. I presume the net area doesn't include newly formed lakes, but I'm not sure.

line 152: talik is defined later in the paper as a thaw bulb, but I think this is the first reference to it.

lines 162-163: is c14 atmospheric monitoring for a permafrost carbon source realistic, given the other sources of old carbon in the atmosphere?

line 176-177: the 27 Tg/yr value for methane is for RCP8.5, which isn't explicitly stated. It differs for RCP4.5.

Figure 2: I think it would be better to show percentage lake area growth rather than a unit like ha/grid cell that might serve to exaggerate the data, especially since the grid cells are pretty large (7.5 x 7.5 km from the caption). The highest lake area growth (20 ha per grid cell) works out to a 0.3 percent increase.

Figure 3: It would be less confusing to add titles to the right axes of a,b (Percent increase in old C emission) and g, h (Percent CPRE increase). Given the multiple things shown on the graphs, it would be a little easier to follow.

Figure 3 caption, lines 225-226: Could you explain what you mean by "Annual emissions before (c-d) and after (e-f) global warming potential calculated." I thought about that one for a while and didn't quite follow what might be meant.

Figure 4: The EM inversion over lake II shows a clear lower boundary, but lake I is not so clear. I

realize from the supplemental discussion that these inversions have lost some resolution through the 3D stitching, lateral constraints, and 3D interpolations, but the boundary for degraded permafrost seems poorly constrained.

lines 483-488: How old are the refrozen thermokarst basins? If they are likely thousands of years old, then I suppose they would emit carbon like a newly thawed lake would. If they are young, though, wouldn't the fossil carbon be depleted during the initial lake formation?

Supplementary Figure 5: The figure looks to have been changed (the RCP8.5 column added?) since the caption was written. The figure shows 6 parts (a through f), but only a through c are mentioned in the caption.

Supplementary Table 2: Please clarify in the caption the distinction between net change (which shows lake area reduction between 1999 and 2014) and gross lake area increase (which shows lake area expansion).

April 30, 2018

Letter to the Referees

The reviewers' comments on our manuscript (NCOMMS-17-32396-T), entitled "21st-century modeled permafrost carbon emissions accelerated by abrupt thaw beneath lakes," were very helpful. We have responded to these comments in the revised manuscript (reviewer comments repeated here in bold) and attempted to address each of their concerns in the following ways:

Referee #1:

1. The peak behavior of thermokarst-lake emissions under RCP8.5, and the fact that RCP4.5 sees a stronger percentage influence of thermokarst lake dynamics on the permafrost carbon feedback, are both tied directly to the relative temperature thresholds and parameterized rates of change for thermokarst lake expansion vs. thermokarst lake drainage. Parameters controlling thermokarst lake drainage should be identified in Table 1 and the uncertainty quantification carried out with a reasonably large range of parameter uncertainties to show what impact these assumptions have on the main conclusions of the study. There should also be more discussion of how the default parameters for this part of the model are derived.

We appreciate the reviewer's request for more transparency in our parameterization of lake expansion vs. drainage and the degree to which it affects our conclusions. In AThaw, modeled thermokarst lake areas determine carbon release. Our thermokarst lake model describes the evolution of new lake areas as a function of projected surface temperature anomalies. The modelled lake areas (F_{TKL}) express net newly formed lake areas, which describe the sum of total lake increase and drainage loss. As there is not a direct AThaw model parameter which controls drainage only, a straight forward sensitivity analysis in the sense that the reviewer may be thinking of is not feasible. Explicit modeling of lake drainage would require the implementation of process-based simulations of the formation of drainage networks and prescription of landscape geomorphology, which are aspects of future development of comprehensive land surface models. In our uncertainty analysis we varied the two key lake parameters that determine net lake evolution: The maximum lake area (F^{TKLmax}) and the optimum temperature anomaly (dT^{TKLmax}). By varying these two parameters, we capture a wide spread of trajectories of lake formation and drainage, which are now shown in full for RCP4.5 and RCP8.5 (new Supplementary Fig. 5).

Our ensemble utilizes a wide range of F^{TKLmax} values (~8% to ~40% increase), depending on soil type (see Methods Sec. 1.3). However, the parameter that most strongly controls the dynamics of thermokarst lake formation and drainage is the temperature optimum, dT^{TKLmax} , which is the temperature at which the maximum lake area occurs. Our selection of dT^{TKLmax} corresponds to high latitude surface air temperatures 4°C to 6°C above pre-industrial. This parameter choice is based on paleoenvironmental evidence of Early Holocene warming by several degrees Celsius in Northern Hemisphere land areas (Kaufman et al., 2004; Velichko et al., 2002; Marcott et al., 2013) which resulted in rapid and intensive thermokarst activity (Walter et al., 2007; Brosius et al., 2012).

We revised the manuscript to explain that this range of dT^{TKLmax} values was also selected to capture the sensitivity of future surface ice-wedge melt to climate warming since ice-wedge melt leads to ground subsidence and water ponding in troughs, which in turn can lead to

thermokarst. Increases in permafrost temperature, which typically mirror increases in air temperature, have been observed in many Arctic regions, with warming of up to 3 °C since the 1970s in some of the coldest permafrost regions (Burn and Kokelj, 2009; Romanovsky et al., 2010, Romanovsky et al. 2017). Permafrost warming often coincides with active layer deepening and increased ground-ice melt. Widespread degradation of surficial ice wedges and a significant increase in areas of water-filled polygonal troughs have been linked to climatic warming during the past few decades in northern Alaska, the Canadian Arctic Archipelago, and Siberia (Jorgenson et al. 2006, Raynolds et al. 2014, Liljedahl et al. 2016). Additional permafrost warming on the order of 2 to 3°C anticipated by 2050 for RCP4.5 and RCP8.5 (Nicolsky et al. 2016), is expected to intensify permafrost and surface ice wedge degradation (Lee et al. 2014), enhancing thermokarst lake expansion and drainage. For RCP4.5, permafrost warming slows beyond 2050 (Nicolsky et al. 2016), supporting AThaw ensembles of lower net thermokarst lake formation during the late 21st century. In contrast, extreme warming after 2050 for RCP8.5 is expected to heavily degrade permafrost, such that near-surface permafrost disappears entirely from many arctic regions. These conditions lend support to AThaw parameterization, which leads to net lake drainage in the later part of the century for RCP8.5 as ice wedges degrade further and drainage networks develop.

While the 4-6°C dT^{TKLmax} range used in our modeling was prescribed based on paleoenvironmental evidence, historical observations, and modeling of future permafrost dynamics, we consider the implications of using smaller or larger dT^{TKLmax} values. Smaller dT^{TKLmax} values would imply that the maximum area of thermokarst lake coverage would occur earlier in the century. This would result in a relatively larger AThaw increase to the PCF earlier in the 21st century and a lower AThaw contribution to PCF toward the end of the century due to more widespread drainage. If dT^{TKLmax} aligned more closely with the range of simulated high latitude warming inferred from higher concentration pathways (e.g. $dT^{TKLmax} > 6$ °C), we would expect maximum thermokarst lake coverage (i.e. F^{TKLmax}) to occur later in the century, less drainage of AThaw lakes during the 21st century, and a larger relative contribution of AThaw to end-of-the century PCF under stronger future warming.

In response to the reviewer's requests for greater transparency in our parameterization of lake expansion vs. drainage, uncertainty quantification based on a wide range of parameter uncertainties, and an assessment of their impact on the main conclusions, we have revised the manuscript in the following ways: We revised the manuscript text, methods section, and Supplementary Table 1 to more clearly identify the parameters that control thermokarst lake expansion and drainage and to show how the default parameters were derived. We added a new Supplementary Fig. 5 to show the full spread in modelled lake trajectories as well as the 68% uncertainty range surrounding the median F_{TKL} values that are the thermokarst lake area basis (net expansion vs. drainage) for our carbon flux modeling in Supplementary Figs. 4 and 5. We added a discussion in the Methods section about the sensitivity of our results to the prescribed dT^{TKLmax} values.

This reviewer also questioned the influence of our AThaw model parameterization and associated uncertainties on our main conclusions, and in particular the conclusion that while permafrost-carbon emissions from lakes are a similar magnitude under RCP4.5 and RCP8.5, the impact of these increased emissions on circumpolar permafrost carbon radiative effect (CPCRE) is more pronounced in the moderate forcing scenario (RCP4.5) compared to the strong (RCP8.5) forcing scenario. In the previous version of the manuscript we failed to highlight that the relative difference in AThaw impact on CPCRE is driven more heavily by the divergence in the earth

system model CLM estimates of terrestrial soil emissions than it is by our parameterization of AThaw emission estimates for RCP4.5 vs. RCP8.5. Within our modeling framework and accounting for a wide span of variability in modeling parameters, AThaw shows a similar magnitude of thermokarst lake emissions under both forcing scenarios despite the peak behavior of AThaw emissions for RCP8.5 (which is due to drainage overwhelming expansion of new lakes at the end of the century) and the lack of this peak behavior for RCP4.5 (since expansion and drainage approach equal rates by 2100). The core reason is that while thermokarst-lake initiation and drainage rates are linked to climate, thermokarst-lake growth – the long process between initiation and drainage that results in most of the carbon release – has dynamics (e.g. talik growth, shore expansion) that once started become rather decoupled from climate due to strong linkage with local factors such as ground ice content and ice-layer thickness (Jorgenson & Shur 2007, Hinkel et al. 2012, Kanevsiy et al. 2014, Edwards et al. 2016). Hence, thermokarst lakes are found across the entire range of Arctic climatic zones and permafrost temperatures as long as there is sufficient ground ice (Burn & Smith 1990, Grosse et al. 2013). A thermokarst lake on the New Siberian Islands has the same potential to release carbon as a thermokarst lake in Central Yakutia; the differences are largely not determined by climate (or RCP conditions) but by local conditions such as permafrost soil carbon and ground ice contents. Hence, either RCP scenario will result in more lakes (earlier or later) and both scenarios will have a similar emission magnitude linked to maximum lake areas, but relative to anthropogenic emissions, the thermokarst lake emissions from RCP4.5 will be more relevant.

The lack of peak behavior caused by late century lake drainage in RCP4.5 has less of an impact on our conclusions. If we were to force peak behavior onto the RCP4.5 AThaw emissions by subtracting a fraction of the emissions after 2070 as a proxy for increased drainage, the relative impact on CPCRE would still exceed that of RCP8.5 due to the much lower CLM gradual thaw emissions for RCP4.5 (Reviewer 1, Fig. R1).

Actual results: RCP4.5 RCP8.5 Forced-peak results: RCP4.5

Fig R1. Impact on conclusions of forcing a peak in RCP4.5 AThaw emissions under the alternative assumption that thermokarst lake drainage exceeds expansion in the late 21st century. While the actual AThaw model shows an increase in lake expansion relative to drainage in the 21st century for RCP4.5, we here show that a forced peak causing drainage to exceed expansion after 2070 for RCP4.5 (an alternative assumption) does not change the four main conclusions (C1-C4) of our study. The conclusions still hold that accounting for abrupt thaw beneath thermokarst lakes during the 21st century significantly increases: cumulative C emissions from permafrost soils (Conclusion 1), the release of old (^{14}C -depleted C) from permafrost soils (Conclusion 2), circumpolar permafrost carbon radiative effect (CPCRE) (Conclusion 3), and the CH_4 contribution to radiative forcing (RF) (Conclusion 4). In all cases, it also still holds that the impact of thermokarst-lake emissions on the permafrost carbon feedback is more pronounced in the moderate forcing scenario (RCP4.5) compared to the strong (RCP8.5) forcing scenario, and this is driven not by AThaw parameterization of late century drainage vs. expansion, but rather by the large difference in CLM terrestrial soil emissions between RCP4.5 (low CLM land emissions) vs. RCP8.5 (high CLM land emissions, see main manuscript).

To improve clarity, we have revised the manuscript to explain that the high AThaw impact on CPCRE under RCP4.5 is caused by differences in the responses of gradual vs. abrupt-thaw dynamics to moderate climate forcing. In the gradual thaw setting for RCP4.5, atmospheric carbon uptake by plants is stimulated more than decomposition of soil organic matter. However, the same degree of warming triggers an acceleration of abrupt thaw via thermokarst-lake formation on a maximum of 4.9% (3.0 to 6.6%, 68% uncertainty range on peak thermokarst area) of the permafrost-dominated landscape in AThaw (revised Supplementary Fig. 4). In contrast, stronger forcing in RCP8.5 accelerates active layer thickening, leading to relatively higher emissions from terrestrial permafrost soils by the end of the century.

Finally, we acknowledge in the revised manuscript that our ability to accurately quantify uncertainties is limited given the use of this single model and its assumptions in a highly complex and large system. Our AThaw modeling results show many similarities to independent modeling of 21st century thermokarst lake development by van Huissteden et al. (2011) and Tan & Zhuang (2015), which are the only other studies we know about on this topic. Despite their entirely different modeling frameworks, these studies show sustained growth of lakes under moderate (RCP4.5) and strong (RCP8.5) forcing conditions until around year 2070. AThaw and van Huissteden's model generally agree in peak lake-area behavior and magnitude of their emissions around 2060 to 2070 followed by a steep decline in areas and emissions invoked by lake drainage for RCP8.5. The two models diverge for RCP4.5 between the period of 2070 and 2100, since during that time, gain in new lake areas exceeds drainage of those new lake areas in AThaw until 2100 when expansion and drainage begin to balance each other, while van Huissteden's model suggests a dominance of drainage commencing at around 2070. Fig. R1 shows that a forced dominance of drainage over expansion in AThaw for RCP4.5 (following the pattern of van Huissteden et al.) would not lead to a significant change in our results or conclusions

Furthermore, our observations of current and historic lake area patterns and emissions support our assumptions that many lakes will continue to expand and new lakes form (even in regions that might experience a net lake area loss due to complete drainage of a smaller number of individual lakes), and that this sustained thaw of permafrost beneath newly formed lake areas will overwhelm the fluxes associated with adjacent areas of lake drainage (see revised Fig. 3, revised main text, and new Supplementary Table 2).

2. From the literature cited, it is not obvious that there is a dominance of one process over the other for historical or present-day conditions, or that, under the currently experienced warming, expansion is overtaking drainage as a signal that can be attributed to anthropogenic climate change.

We agree with the reviewer that there is no obvious dominance of one process over another (expansion vs. drainage) for historical or present-day conditions as a signal that can be attributed to anthropogenic climate change. In the Holocene record there is evidence for rapid and intense thermokarst-lake formation (Walter et al. 2007, Brosius et al. 2012) in response to early Holocene warming by a few degrees Celsius in Northern Hemisphere land areas (Kaufman et al. 2004, Velichko et al. 2002, Marcott et al. 2013). Paleoenvironmental records suggest that these lakes persisted for several thousand years before draining (Walter Anthony et al. 2014). Lake formation rates slowed in the late Holocene due to cooler climate conditions and formation of peat, which provides thermal protection against ground ice-melt. During historical and present

times, lake changes are quite diversified at pan-arctic and regional scales (See new Supplementary Table 2d). In some regions, large (e.g. +48%) net increases in lake area are observed driven by strong precipitation increases. In many regions, there is a small net decrease in lake area; however, resolution of remote-sensing base images also impacts results since the loss of lake area affecting numerous contiguous pixels is readily detected in coarse-resolution imagery (e.g. 30-m to 120-m resolution), while formation of small, new lakes can only be detected with fine resolution (e.g. 1-m) imagery.

We have revised Supplementary Fig. 3 to show that formation of new thermokarst lakes during recent decades goes unnoticed in Landsat analyses covering a short time period at ≥ 30 m resolution; however, this process of new lake formation and growth in permafrost uplands is likely to accelerate in a warmer climate (Jorgenson et al. 2006, Reynolds et al. 2014, Nicolsky et al. 2016). The important question is whether current upland geomorphology can sustain lateral growth of new lakes to persist through the 21st century on valley-fragmented uplands. Based on average expansion rates observed with high resolution remote sensing data during the past 60 years (0.3 m yr^{-1}) and proximity of newly formed lakes to drainage channels (Jones et al. 2011), most of these newly formed lakes are likely to survive this century. In a much warmer Arctic (RCP4.5 and RCP8.5), not just the flat uplands will thaw, but also the older drained lake basins containing lots of ground ice now protected by a layer of peat. Hence, in a much warmer and moister Arctic, we will likely see new lakes forming in older drained lake basins and remnant lakes rapidly expanding.

How likely these lakes are to drain depends on local factors such as ice content, permafrost thickness, and proximity to drainage channels). Nitze et al. (2017) observed a slight net decrease in many regions of the Arctic, driven by the drainages of a small number of very large single lakes, where single events have a very big impact on the overall lake area balance. However, those lakes are not likely to be big CH₄ emitters today since they will have depleted their supply of labile C in thawed permafrost earlier in their development thousands of years ago. It is the talik formation of new small lakes, as well as talik expansion of existing lakes thawing and mobilizing old permafrost sediments, that fuels CH₄-producing microbes. This process of gross lake area growth occurs in the majority of lakes on the landscape (Jones et al. 2011, Grosse et al. 2013, Kokelj & Jorgenson et al. 2013), strongly counterbalancing gross lake area losses.

In the revised manuscript we have included a new set of supplementary tables (Supplementary Tables 2a-d) to more comprehensively summarize the literature on historic and present-day lake area change dynamics. We have also applied field-work observed carbon flux factors associated with gross lake area losses (GLAL) and gross lake area gains (GLAG) to our own lake change analysis of 73,804 lakes in Alaska, to show that despite regional net lake area loss, a region will still exhibit a net increase in carbon emissions associated with lake change since carbon fluxes on the drained portion of the landscape are small in comparison to fluxes associated with lake-area expansion that taps into the permafrost soil carbon pool (Supplementary Table 3). Finally, we show that until GLAL area is at least 4 and 7 times GLAG area, a ratio yet to be observed in remote-sensing records (Supplementary Table 2b, d), any GLAG will result in a net increase in regional lake carbon emissions, and specifically the release of old carbon.

3. Lines 55-65: would the mechanisms described here also apply to water accumulation in polygonal tundra, at spatial scales smaller than what would normally be considered a lake?

Yes, the mechanisms of water ponding leading to accelerated, deeper thaw and mobilization of old carbon would apply at smaller spatial scales, such as water ponding in troughs (this is indeed a common initial stage of thermokarst lake formation! – see Kokelj & Jorgenson 2013); however, it only applies in cases where thawed ground does not seasonally refreeze allowing formation of a talik. To improve clarity, we have revised the manuscript to explain that, “Water pooling in collapsed areas leading to formation of taliks (unfrozen thaw bulbs), accelerates permafrost thaw far faster and deeper (Fig. 1a) than would be predicted from changes in air temperature alone (Kessler et al. 2012, Arp et al. 2016, Langer et al. 2016).”

4. Fig S3: Are there no examples in this region of lake contraction, due to lake drainage or other processes? I was surprised that the scale bar shows only lake expansion fraction.

Based on the importance of gross lake area expansion to mobilization of old permafrost carbon stocks we had focused the original manuscript on gross lake area gains. However, lake area losses certainly also occur in the study regions. We have revised the manuscript text, Fig. 3, new Supplementary Table 2, Supplementary Table 3, and Supplementary Fig. 3 to show the full spectrum of gross lake area loss, gross lake area gain, and net lake area change, as well as the carbon flux implications of these lake changes. See also Reviewer 1, comment 2.

5. Lines 190-192: The statement that, once formed, abrupt thaw features continue to develop even under shifts to a cooler climate, seems to leave out consideration of the potential for lake drainage.

We appreciate this comment and have revised the main manuscript text to more comprehensively describe processes that lead to lake area loss and implications for carbon cycling. We explain many newly formed lakes will ultimately be subject to drainage (Schneider von Deimling et al. 2016, van Huissteden et al. 2011, Tan & Zhuang 2015) when they intersect topographical drainage gradients by lateral expansion (Morgenstern et al. 2011), from elevated water levels (Jones & Arp 2015), or when taliks penetrate permafrost, allowing the potential for internal drainage to the groundwater system (Yoshikawa & Hinzman 2003) (Supplementary Figs. 4, 5). However, models consistently predict that a net drainage of newly formed lakes will not occur before the late 21st century (Schneider Von Deimling et al. 2016, van Huissteden et al. 2011, Tan & Zhuang 2015) (Supplementary Figs. 4, 5), at which time gradual permafrost thaw begins to accelerate (Koven et al. 2015, McGuire et al. 2018), further intensifying the PCF to climate warming. Furthermore, it is conceivable that for RCP8.5, when temperatures are warm enough to prevent refreezing of taliks following lake drainage (Nicolsky et al. 2016) that CH₄ fluxes from drained basins will be higher than they are today, since in today’s cooler climate, permafrost aggradation typically follows lake drainage. Methanotrophy (Whalen & Reeburg 1996) will offset emissions of CH₄ produced in late 21st-century drained-lake-basin taliks. However, ecosystem-scale microbial studies show a higher temperature response by methanogenesis than by methanotrophy or by CO₂ fluxes attributable to respiration and photosynthesis (Yvon-Durocher et al. 2014, Sepulveda-Jauregui et al. 2018). This indicates that in a warmer world, CH₄ emissions and the ratio of CH₄ to CO₂ emissions from individual ecosystems will (Yvon-Durocher et al. 2014).

Referee #2:

1. This is an impressive study focused on predicting future greenhouse gas release resulting from abrupt thawing of permafrost beneath thermokarst lakes. To my knowledge, a number of recent studies have pointed towards thermokarst areas being particularly significant for permafrost carbon feedback. But this is perhaps the first study to attempt to quantify the emissions, which is largely a modeling exercise using a combination of field measurements, remote sensing and radiocarbon analysis. I think there will be a lot of interest in this study, and it will strongly influence research into permafrost carbon feedback.

I have been asked to comment on the radiocarbon analyses in this study which represent a relatively minor part of the study. The radiocarbon analyses undertaken for this study were all performed on samples of methane-rich gas (radiocarbon measurements of carbon dioxide are reported from a published study (Fig. 1c)). The methods used to measure the samples are reasonably routine and the description in the Supplementary Information (SI) indicates that care was taken during the collection, preparation and radiocarbon measurement of these samples. I therefore have high confidence in the measurements.

I think it is a pity that radiocarbon measurements of carbon dioxide were not undertaken as part of this study. I appreciate that the CO₂ concentration in the samples was much lower than methane, and less important in terms of radiative forcing. But I think some analyses of CO₂ would have improved the study. As it is, Fig 1c seems odd, because in this ¹⁴C comparison between gradual-thaw sites and abrupt-thaw sites the former are for CO₂ and the latter are for CH₄. Therefore it's not a like-for-like comparison because the terrestrial CO₂ will have a much greater influence of modern carbon from plants (e.g. Schurr et al. measured ecosystem respiration). If CH₄ had been dated at the terrestrial sites it is very possible that it would have been even older than the terrestrial CO₂.

We thank the reviewer for their suggestion to show like for like data in Fig. 1c. Based on this comment, we contacted the authors of the terrestrial active layer studies (Drs. Ted Schuur and Claudia Czimczik) to inquire about the availability of any ¹⁴C-CH₄ ages they might have. Schuur responded that they have just started to measure CH₄ fluxes at Healy, Alaska but have not done any radiocarbon dating yet on CH₄. In northwest, Greenland, Czimczik's group did not have the opportunity to measure CH₄ age because they observed CH₄ uptake only. There, the soils are very rocky and well drained. Standing water occurs during snowmelt, but at that point the soil is frozen and the meltwater is basically at 0 °C in a shallow film above the soil in close contact with the air, so there was not CH₄ production.

To complement the ¹⁴C-CO₂ of the terrestrial active-layer studies, we have added to the revised manuscript our own ¹⁴C-dated CO₂ from some of the lake ebullition bubbles for which ¹⁴C-CH₄ ages are already shown. Due to the low concentrations of CO₂ in lake bubbles (typically <2% by volume) and a previous focus on CH₄ measurements, our ¹⁴C-CO₂ data are sparse. We found that CO₂ ages in bubbles were younger than those of CH₄, a trend that has also been observed in dissolved gases in arctic lakes (Elder et al. 2018). While the differences in ages are not well understood, we acknowledge that if CH₄ had been measured on the active layer sites, it also may have been older than CO₂ ages. However, Cooper et al. showed that CH₄ from

terrestrial sites may not be very old. They reported, “The results from these two contrasting sites in different permafrost zones demonstrate that, where substantial CH₄ fluxes occurred, they were dominated by anaerobic decomposition of recent C inputs. Total rates of CH₄ release from previously-frozen C were low irrespective of differences in time since thaw, vegetation community composition and/or water-table depth.” We interpret the much older ¹⁴C ages of gases emitted from the lakes’ thermokarst expansion zones as the result of abrupt thaw that extends into deep (8 to 15 m), old permafrost faster than gradual thaw in terrestrial active layers, whose thickness is only 0.5 to 0.6 m. Since permafrost soil carbon typically increases in ¹⁴C age with depth (Meyer et al. 2008, Murton et al. 2015, Walter Anthony et al. 2016), decomposition at increasingly greater depths should produce carbon gases with older ¹⁴C ages.

2. Some of the radiocarbon concentrations appear to be very close to the limit of detection for ¹⁴C, which to me raises the question of whether the methane is derived from the breakdown of organic material in the permafrost, or whether it could be thermogenic. If there is likely to be organic matter that is >40,000 years old then the latter explanation is not required. However, I think a contribution from a thermogenic source would have implications for the interpretation of the methane fluxes. For example, the observation is reported that methane emissions in older thaw lakes are lower than in younger thaw lakes because the labile fraction of older lakes has been mineralised. Could an alternative interpretation be that much of the methane is thermogenic, built up over time and trapped in the permafrost, and that thawing just allows its release (explaining the extremely low ¹⁴C concentrations)? How would this effect the model predictions?

We thank the reviewer for their thoughtful consideration of alternative explanations for the highly ¹⁴C-depleted values. In fact, we have observed thermogenic CH₄ sources in other arctic lakes; but we did not observe them in any lakes in the current manuscript. Walter Anthony et al. (2012) distinguished geologic vs. ecological methane from a large number of lakes in Alaska and Greenland using ¹⁴C, stable isotopes δ¹³C-CH₄, δD-CH₄, Bernard ratios, fluxes, geospatial relationships, and comparisons to gas well data sets. The lake bubble δ¹³C-CH₄ values observed in this study suggest a microbial CH₄ origin [δ¹³C-CH₄ -69 ± 5 (‰), -55 to -82 (‰) min to max]. The reviewer is correct that the ¹⁴C-CH₄ ages around 40,000 years correspond to similarly old yedoma permafrost soil organic carbon (SOC) ages surrounding lakes (see Walter Anthony et al. 2016). We have revised the manuscript by adding δ¹³C-CH₄ data to the new Supplementary Table 5 and by clarifying in the text the microbial origin of methane and the similarities between ¹⁴C-CH₄ ages and ¹⁴C-SOC ages.

3. Figure 1 c. There are 11 data-points for methane radiocarbon concentrations plotted. In the SI (p 8) it says that “72 ebullition events from 11 thermokarst lakes ..were dated”. Were individual ¹⁴C analyses undertaken on each ebullition event and the results combined, or were only 11 samples analysed by combining the ebullition events for each lake? Some clarification of this would be helpful.

We have revised the figure caption to explain that each point in Fig. 1c represents the mean for a study site. To improve clarity, we have added error bars to the figure and report ¹⁴C ages of individual events in a new Supplementary Table 5.

4. I think it would be valuable to other researchers if the individual radiocarbon results

were reported in the SI, along with any ancillary measurements such as %CH₄ and delta 13C. If delta 13C was measured then the results should inform on whether there was any thermogenic methane or not.

We have also followed the reviewer's suggestion to provide the original ¹⁴C and δ¹³C data in a Supplementary Table (new Supplementary Table 5).

5. Figure 1c. Left-hand-side y-axis title should I believe be Fraction modern carbon (rather than Percent modern carbon) – otherwise the ages would all be old and the right-hand-side x-axis wouldn't be correct.

We thank the reviewer for this correction and have revised the axis title to read 'Fraction modern carbon.'

Referee #3:

1. With only general knowledge of carbon emissions related to permafrost degradation, I learned quite a bit reading this interesting, concise (at least the main part), well written, and important paper. The authors make a strong argument for the role abrupt-thaw permafrost beneath predicted new and expanding thermokarst lakes will play in increasing greenhouse gas concentrations as the climate warms, and suggest that these contributions are large enough to be included in better models of greenhouse gas trajectories through the 21st century. Although principally a modeling effort (and I would quibble about the title lacking any mention that this is a synthetic study), the authors include large amounts of field data, some relevant remote sensing, and even some airborne geophysics to the mix. As a result, I have no major complaints about the paper and feel it would be a worthy addition to the permafrost and climate-change literature. I guess the greatest amount of heartburn from the paper is the indiscriminate mixing of reality and model results; we all recognize models have their limitations and the future is unlikely to unfold as the authors might predict. Some effort to more clearly emphasize that, starting with the title, might be appropriate.

We appreciate the reviewer's comments. We have added 'modeled' to the title and have carefully revised the text to more clearly distinguish model results from observational results.

2. Also, it seems that most of the model-based projections differ greatly from recent and historical experience, and those effects begin rather soon. Why have we not seen strong beginnings of these effects in recent decades, or if we have, emphasizing those would help bridge the belief gap between experience and model projections.

The reviewer raises good points. In Walter Anthony et al. (2016) we quantified CH₄ emissions associated with pan-arctic scale thermokarst-lake formation and expansion during the past 60 years and found that while emissions within individual lakes was proportional to the eroded soil carbon stocks adjacent to lakes, the dramatic increase in permafrost carbon emissions that is expected to imminently occur (based on modeling) shows no sign of having commenced (based on field work and remote sensing observations) (See Fig. R3). It seems we are on the brink of this abrupt change.

Fig R3. Modeled permafrost carbon emissions during the Holocene and this century. (modified from Walter Anthony et al. 2016, Fig. 3).

To improve understanding of this critical inflection point (little change in thermokarst emissions during the past 60 to 8,000 years vs. huge increase during the next 80 years), we have revised the current manuscript by adding a synthesis of additional relevant literature and a new Supplementary Table 2d. Specifically, “In the context of Holocene-scale thermokarst dynamics, present-day AThaw CH_4 emissions (0.7 to 4.0 Tg yr^{-1} , 68% range) represent no significant change from thermokarst-lake emissions over the past 8,000 years (Walter et al. 2007), a pattern that is also consistent with no significant changes in arctic natural CH_4 sources during the historical record of atmospheric monitoring (AMAP 2015)... Widespread acceleration of gross lake area gain during recent decades has yet to be observed among studies of multitemporal satellite imagery (Romanovsky et al. 2017); however conclusive evidence requires quantification of gross lake area growth using high-resolution imagery in multiple time slices, an analytical combination that is rare in the literature (e.g. Jones et al. 2011, Necsoiu et al. 2013) (Supplementary Table 2d). Analyses of aerial photographs revealed that surface ice-wedge melt, a critical first step in thermokarst-lake formation (Shur & Osterkamp 2007), abruptly increased during the last 30 years in several pan-arctic areas in response to exceptionally warm summers and a long-term upward trend in summer temperature (Raynolds et al. 2014, Jorgenson et al. 2006, Liljedahl et al. 2016). Other studies demonstrated the recent transition from bedfast to floating lake ice in Arctic Alaska (Arp et al. 2016). Diminishing lake ice will also influence this inflection once many more shallow lakes undergo CH_4 -producing talik development. Terrestrial Arctic warming of $4\text{-}6 \text{ }^\circ\text{C}$ (RCP4.5) and $>7 \text{ }^\circ\text{C}$ (RCP8.5) projected to occur this century (IPCC 2013, McGuire et al. 2018) will be unprecedented for the Holocene (Kaufmann et al. 2004, Marsicek et al. 2018) and is anticipated to greatly increase both gross lake area growth (Schneider von Deimling et al. 2015, van Huissteden et al. 2011, Tan & Zhuang 2015) and the PCF (Schuur et al. 2015, Schaefer et al. 2014, Koven et al. 2015a, Koven et al. 2015b, Burke et al. 2017).”

Being a non-expert on the principal focus of the paper, I can't find much fault with the approach, results, or interpretations. There are a few places where more clarity would be helpful to a general scientific audience, and there are embarrassingly few other issues I found. The ones I saw are listed below by line number or page number for the supplemental material.

Abstract: Well written and concise.

N/A.

3. line 30: Given that the RCPs are concentration pathways, would it be better to say moderate concentration rather than moderate warming? We're sure about the concentration path, but I think the warming magnitude is more uncertain.

RCP4.5 and RCP8.5 are pathways that would result in preindustrial to 2100 radiative forcing being 4.5 W m^{-2} ($\sim 650 \text{ CO}_2$ equivalent) and 8.5 W m^{-2} ($\sim 1,370 \text{ CO}_2$ equivalent), respectively (van Vuuren et al. 2011). To achieve the RCP4.5 pathway would require carbon emissions per energy consumption by global human society to decrease by 75% during this century (van Vuuren et al. 2011), while the RCP8.5 assumes little or no mitigation. Terms frequently encountered in the literature to describe these pathways or scenarios include, “warming pathway, forcing pathway, climate change pathway, mitigation pathway, stabilization/non-stabilization pathway.” We opted to retain “warming scenario” because it makes more intuitive sense to a broad audience that permafrost thaw would respond to climate warming than it would to CO_2 equivalent concentration increases. We also opted to retain this phrasing because the concentration scenarios the reviewer mentioned are also subject to uncertainty as they depend on the realism of underlying carbon cycle models.

4. lines 103-108: Here's one place where the Holocene record to 8 ka and the historic record show no apparent acceleration in thermokarst dynamics, yet the models predict those will begin "real soon now." 1.2 to 1.8 percent gross lake area growth since 1999 doesn't seem to be a strong argument for acceleration either, although the authors follow that with much larger gross growth rates in other Arctic areas. Please also clarify what you mean by gross lake area growth; the net growth shown in Table S2 shows lake-area reductions. I presume the net area doesn't include newly formed lakes, but I'm not sure.

To improve clarity, we have revised the text to explain that Lacking a longer observational record dissected into multiple time slices and higher resolution imagery analyses we cannot discern this gross lake area increase as different from natural thermokarst-lake processes that occur irrespective of climate warming (Jones et al. 2011, Jorgenson et al. 2010, Grosse et al. 2013). Our observed 1999-2014 gross lake area gain (154 km^2) is outweighed by gross lake area loss (i.e. lake drainage, 330 km^2) for the same study extent ($12,798 \text{ km}^2$ total lake area; Fig. 3). However, this lake change dynamic still contributes and additional $0.9 \text{ Tg C-CO}_2\text{e}$ of landscape-scale carbon emissions to the atmosphere over the 15-yr study period when field-measured fluxes are applied to increasing and decreasing lake-area changes (Supplementary Table 3). Our net carbon emission estimate is likely conservative because the 30-m resolution Landsat-based analysis did not account for the formation and growth of numerous smaller lakes that are only detectable with finer-resolution imagery (Supplementary Fig. 3). Despite net lake area loss, landscape-scale carbon emission to the atmosphere remains positive because conversion of upland terrestrial ecosystems with relatively low carbon fluxes (Supplementary

Table 2b) to newly formed thermokarst-lake areas with high CH₄ emissions results in a 130- to 430-fold increase in emissions per square meter of land surface change (Supplementary Table 2c). The contrasting drainage of lower-emitting older portions of lakes and the establishment of productive, wetland vegetation in drained lake basins leads to smaller changes in carbon fluxes (factor of -0.004 to +0.08).

In addition to revising the text, we added gross lake area loss values to Supplementary Table 3, modified Supplementary Figure 3 to show the importance of high-resolution imagery in quantifying gross lake area gain, and added an entirely new set of Supplementary tables (Supplementary Table 2) to synthesize pan-arctic studies of gross lake area gains and losses and their carbon flux implications.

5. line 152: talik is defined later in the paper as a thaw bulb, but I think this is the first reference to it.

We thank the reviewer for pointing this out. We revised the text to define talik at its first reference, which is now in the Introduction.

6. lines 162-163: is c14 atmospheric monitoring for a permafrost carbon source realistic, given the other sources of old carbon in the atmosphere?

The reviewer makes a good point that the ¹⁴C-depleted signature of methane emitted from thermokarst expansion zones may not be detectable given other sources of old carbon in the atmosphere. We agree that the estimated present-day ~4 Tg CH₄ yr⁻¹ emitted from pan-arctic lake expansion zones is unlikely to be detected by atmospheric ¹⁴C measurements amidst other natural and anthropogenic fossil methane sources, which are thought to be 150-180 Tg yr⁻¹ (Lassey et al. 2007). However, an increase in ¹⁴C-depleted CH₄ sources from thermokarst on the order of 20 to 27 Tg CH₄ yr⁻¹ this century (RCP4.5 and RCP8.5 respectively) may make detecting the permafrost carbon source more feasible.

7. line 176-177: the 27 Tg/yr value for methane is for RCP8.5, which isn't explicitly stated. It differs for RCP4.5.

We have now clarified in the text that the 27 Tg yr⁻¹ increase in CH₄ emissions is for RCP8.5.

8. Figure 2: I think it would be better to show percentage lake area growth rather than a unit like ha/grid cell that might serve to exaggerate the data, especially since the grid cells are pretty large (7.5 x 7.5 km from the caption). The highest lake area growth (20 ha per grid cell) works out to a 0.3 percent increase.

We revised this figure (now Fig. 3) to show percentage gross lake area growth as suggested. We also added new panels to show percentage gross lake area loss, net lake area change, and net change in carbon fluxes.

9. Figure 3: It would be less confusing to add titles to the right axes of a,b (Percent increase in old C emission) and g, h (Percent CPRE increase). Given the multiple things shown on the graphs, it would be a little easier to follow.

We thank the reviewer for this helpful suggestion and have added titles to the axes (now Fig. 4).

10. Figure 3 caption, lines 225-226: Could you explain what you mean by "Annual emissions before (c-d) and after (e-f) global warming potential calculated." I thought about that one for a while and didn't quite follow what might be meant.

To improve clarity, we revised this sentence to read, "Annual CH₄ and CO₂ emissions expressed as Tg yr⁻¹ (c-d) and collectively as Tg C-CO₂e yr⁻¹ (e-f) based on a GWP₁₀₀ of 28 (Myhre et al. 2013) and units conversions shown in Supplementary Table 2a."

11. Figure 4: The EM inversion over lake II shows a clear lower boundary, but lake I is not so clear. I realize from the supplemental discussion that these inversions have lost some resolution through the 3D stitching, lateral constraints, and 3D interpolations, but the boundary for degraded permafrost seems poorly constrained.

We appreciate this comment. The apparent difference in thaw depth interpretation for Lake I and Lake II is an artifact of the modeling. Supplementary Fig. 7 1-D resistivity models (EM1DFM) created from preliminary data collected directly along the flight lines show a more consistent relationship between resistivity and our interpretation of thaw depth for the two lakes. In the revised figure (now Fig. 5) legend we have directed the reader to Supplementary Information where we have added an explanation for the apparent differences in thaw boundary interpretations between lakes I and II. In the modeling process, the thicker the upper conductive area (thawed area) the more difficult it is to see a resistive (frozen area) area below it. If a resistive area exists it may show higher resistivity in the modeling process than an area with similar physical properties and no conductive layer above it. There is an increase in the resistivity in the model with depth under Lake I suggesting a decrease in free water and temperature. Another way to view the interpretation is Lake I has a thermal footprint laterally and below the actual thawed area and a much larger footprint than Lake II; the first 30 meter subsurface below Lake I is different than the subsurface 30 to 100 meters in depth. The model also supports that the frozen material is likely colder and thicker below lake II than lake I.

12. lines 483-488: How old are the refrozen thermokarst basins? If they are likely thousands of years old, then I suppose they would emit carbon like a newly thawed lake would. If they are young, though, wouldn't the fossil carbon be depleted during the initial lake formation?

As the reviewer alludes, carbon stocks and quality (bioavailability) depend on basin age and size. In many cases (e.g. centers of large basins), the labile fraction of the original yedoma carbon stocks were depleted during the initial lake formation (i.e. first-generation lake). However, labile Pleistocene-aged yedoma remains under a smaller areal fraction of drained lake basins, in particular along the margins of large lakes and beneath smaller lakes that did not completely thaw through the Pleistocene-aged yedoma permafrost soil profile before the lakes drained. As the reviewer suggests, time since drainage is another factor to consider, since peat accumulates in drained lake basins (Jones et al. 2012). However, our previous work (Walter Anthony et al. 2014) showed the largest carbon accumulation rates occur in mature lakes prior to drainage. Our AThaw emissions from lakes forming in refrozen thermokarst basins rely on carbon stocks and ground ice contents measured in a large number of basins spanning the vast Yedoma region in Siberia and Alaska (Strauss et al. 2013, Walter Anthony et al. 2014). Observed basins varied in age and lake formation stage and are assumed to represent landscape scale variability. We have referenced these large geospatial data sets in our revised manuscript.

13. Supplementary Figure 5: The figure looks to have been changed (the RCP8.5 column added?) since the caption was written. The figure shows 6 parts (a through f), but only a through c are mentioned in the caption.

We thank the reviewer for this correction. We revised this figure (now Supplementary Fig. 6) to accurately reference all six panels.

14. Supplementary Table 2: Please clarify in the caption the distinction between net change (which shows lake area reduction between 1999 and 2014) and gross lake area increase (which shows lake area expansion).

We have added to this Supplementary Table an additional row showing gross lake area loss and clarified in the caption that net lake change is the sum of gross lake area loss and gross lake area gain.

We thank the reviewers for their helpful remarks, which have caused us to improve the paper. We hope that the revised manuscript will now be found suitable for publication in Nature Communications.

Additional References:

- AMAP Assessment 2015: Methane as an Arctic climate forcer. Arctic Monitoring and Assessment Programme (AMAP), Oslo, Norway. vii + 139 pp (2015).
- Arp *et al.* Threshold sensitivity of shallow Arctic lakes and sublake permafrost to changing winter climate. *Geophys. Res. Lett.* **43**, 6358–6365 (2016).
- Brosius, L. S. *et al.* Using the deuterium isotope composition of permafrost meltwater to constrain thermokarst lake contributions to atmospheric methane during the last deglaciation. *J. Geophys. Res. Biogeosci.* **117**, G01022, doi:10.1029/2011JG001810 (2012).
- Burke, E. J. *et al.* Quantifying uncertainties of permafrost carbon-climate feedbacks. *Biogeosci.* **14**, 3051-3066 (2017).
- Burn, C. R., & M. W. Smith. Development of thermokarst lakes during the Holocene at sites near Mayo, Yukon Territory. *Permafr. Periglac. Process.* **1**, 161-175 (1990).
- Burn, C.R., & Kokelj, S.V. The environment and permafrost of the Mackenzie Delta area. *Permafr. Periglac. Process.* **20**, 83-105 (2009).
- Cooper, M. D. A. *et al.* Limited contribution of permafrost carbon to methane release from thawing peatlands. *Nat. Clim. Change* **7**, 507-511 (2017).
- Edwards, M., Grosse, G., Jones, B. M., McDowell, P. The evolution of a thermokarst-lake landscape: Late Quaternary permafrost degradation and stabilization in interior Alaska. *Sedimentary Geology* **340**, 3-14 (2016).
- Elder *et al.* Greenhouse gas emissions from diverse Arctic Alaskan lakes are dominated by young carbon. *Nat. Clim. Change* **8**, doi.org/10.1038/s41558-017-0066-9 (2018).
- Grosse G., Jones B., & Arp C. Thermokarst Lakes, Drainage, and Drained Basins. In: John F. Shroder (ed.) *Treatise on Geomorphology*, Volume 8, pp. 325-353. San Diego: Academic Press. (2013).

- Hinkel, K. M. et al.. Thermokarst Lakes on the Arctic Coastal Plain of Alaska: Geomorphic Controls on Bathymetry. *Permafr. Periglac. Process.* **23**, 218-230 (2012).
- IPCC in Climate Change 2013: The Physical Science Basis. Contribution of Working Group I to the Fifth Assessment Report of the Intergovernmental Panel on Climate Change (eds Stocker, T. F. et al.) 1535 (Cambridge Univ. Press, 2013).
- Jones, B. M. et al. Modern thermokarst lake dynamics in the continuous permafrost zone, northern Seward Peninsula, Alaska. *J. Geophys. Res. Biogeosci.* **116**, G00M03 (2011).
- Jones, M. C., Grosse, G., Jones, B. M., & Walter Anthony, K. M. Peat accumulation in a thermokarst-affected landscape in continuous ice-rich permafrost, Seward Peninsula, Alaska. *J. Geophys. Res.* **117**, G00M07. doi:10.1029/2011JG001766 (2012).
- Jones, B. & Arp, C. Observing a catastrophic thermokarst lake drainage in northern Alaska. *Permafr. Periglac. Process.* **26**, 119-128 (2015).
- Jorgenson, M. T., Shur, Y. L., & Pullman, E. R. Abrupt increase in permafrost degradation in Arctic Alaska. *Geophys. Res. Lett.* **33**, doi:10.1029/2005GL024960 (2006).
- Jorgenson, M.T., & Shur, Y. Evolution of lakes and basins in northern Alaska and discussion of the thaw lake cycle. *J. Geophys. Res.* **112**, F02S17 (2007).
- Jorgenson, M. T. et al. Resilience and vulnerability of permafrost to climate change. *Can. J. For. Res.* **40**, 1219–1236 (2010).
- Kanevskiy, M. et al. Cryostratigraphy and Permafrost Evolution in the Lacustrine Lowlands of West-Central Alaska. *Permafr. Periglac. Process.* **25**, 14-34 (2014).
- Kaufman, D. et al. Holocene thermal maximum in the western Arctic (0-180oW). *Quat. Sci. Rev.* **23**, 529-560 (2004).
- Kessler, M. A., Plug, L., & Walter Anthony, K. Simulating the decadal to millennial scale dynamics of morphology and sequestered carbon mobilization of two thermokarst lakes in N.W. Alaska. *J. Geophys. Res. Biogeosci.* **117**, doi:10.1029/2011JG001796 (2012).
- Kokelj, S. V. & Jorgenson, M. T. Advances in thermokarst research. *Permafr. Periglac. Process.* **24**, 108-119 (2013).
- Koven, C. D., Lawrence, D. M., & Riley, W. J. Permafrost carbon-climate feedback is sensitive to deep soil carbon decomposability but not deep soil nitrogen dynamics. *PNAS* **112**, 3752-3757 (2015a).
- Koven, C. D. et al. A simplified, data-constrained approach to estimate the permafrost carbon-climate feedback. *Phil. Trans. R. Soc. A.* **373**, DOI: 10.1098/rsta.2014.0423 (2015b).
- Langer, M. et al. Rapid degradation of permafrost underneath waterbodies in tundra landscapes—Toward a representation of thermokarst in land surface models. *JGR Earth Surface* **121**, 2446-2470 (2016).
- Lassey, K. R., Lowe, D. C., & Smith, A. M. The atmospheric cycling of radiomethane and the "fossil fraction" of the methane source. *Atmos. Chem. Phys.* **7**, 2141 - 2149 (2007).
- Lee, H., Swenson, S. C., Slater, A. G., & Lawrence, D. M. Effects of excess ground ice on projections of permafrost in a warming climate. *Environ. Res. Lett.* **9** doi:10.1088/1748-9326/9/12/124006 (2014).
- Liljedahl, A. K. et al. Pan-Arctic ice-wedge degradation in warming permafrost and its influence on tundra hydrology. *Nature Geosci.* **9**, 312-318 (2016).
- Marcott, S. A., Shakun, J. D., Clark, P. U., & Mix, A. C. A Reconstruction of Regional and Global Temperature for the Past 11,300 Years. *Science* **339**, 1198–1201 (2013).

- McGuire, D. A. et al. Dependence of the evolution of carbon dynamics in the northern permafrost region on the trajectory of climate change. *PNAS*, doi.org/10.1073/pnas.1719903115 (2018).
- Meyer, H., Yoshikawa, K., Schirrmeister, L., & Andreev, A. The Vault Creek Tunnel (Fairbanks Region, Alaska): A Late Quaternary Palaeoenvironmental Permafrost Record in Proceedings of the Ninth International Conference on Permafrost (eds. Kane, D. L. & Hinkel, K. M.) pp. 1191-1196 (Institute of Northern Engineering, Fairbanks, 2008).
- Morgenstern, A., Grosse, G., Guenther, F., & Schirrmeister, L. Spatial analyses of thermokarst lakes and basins in Yedoma landscapes of the Lena Delta. *Cryosph.* **5**, 849-867 (2011).
- Murton, J. B. et al. Palaeoenvironmental Interpretation of Yedoma Silt (Ice Complex) Deposition as Cold-Climate Loess, Duvanny Yar, Northeast Siberia. *Permafrost Periglac.* **26**, 208–288 (2015).
- Myhre, G. et al. In: *Climate Change 2013: The Physical Science Basis. Contribution of Working Group I to the Fifth Assessment Report of the Intergovernmental Panel on Climate Change* [Stocker, T.F., D. Qin, G.-K. Plattner, M. Tignor, S.K. Allen, J. Boschung, A. Nauels, Y. Xia, V. Bex and P.M. Midgley (eds.)]. Cambridge University Press, Cambridge, UK and New York, NY, USA (2013).
- Nicolosky, D. J., Romanovsky, V. E., Panda, S. K., Marchenko, S. S. & Muskett, R. R. Applicability of the ecosystem type approach to model permafrost dynamics across the Alaska North Slope. *J. Geophys. Res. Earth Surf.* **121**, doi:10.1002/2016JF003852 (2016).
- Nitze, I. et al. Landsat-based trend analysis of lake dynamics across northern permafrost regions. *Rem. Sens.* doi.org/10.3390/rs9070640 (2017).
- Raynolds, M. K. et al. Cumulative geocological effects of 62 years of infrastructure and climate change in ice-rich permafrost landscapes, Prudhoe Bay Oilfield, Alaska. *Glob. Change Biol.* **20**, 1211-1224 (2014).
- Romanovsky, V., Smith, S., & Christiansen, H. Permafrost thermal state in the polar Northern Hemisphere during the International Polar Year 2007–2009: A synthesis. *Permafrost Periglac.* **21**, 106–116 (2010).
- Romanovsky, V. et al. In: *Snow, Water, Ice and Permafrost in the Arctic (SWIPA) 2017*. Pp. 65-102. Arctic Monitoring Assessment Programme (AMAP), Oslo, Norway (2017).
- Schaefer, K., Lantuit, H., Romanovsky, V. E., Schuur, E. A. G. & Witt, R. The impact of the permafrost carbon feedback on global climate. *Environ. Res. Lett.* **9**, 085003 (2014).
- Schneider von Deimling, T. et al. Observation-based modelling of permafrost carbon fluxes with accounting for deep carbon deposits and thermokarst activity. *Biogeosci.* **12**, 3469-3488 (2015).
- Schuur, E. A. G. et al. Climate change and the permafrost carbon feedback. *Nature* **520**, 171-179 (2015).
- Shur, Y., & Osterkamp, T. E. Thermokarst. Report No. INE0611, Institute of Northern Engineering, University of Alaska Fairbanks, Fairbanks, AK, USA (2007).
- Strauss, J. et al. The deep permafrost carbon pool of the Yedoma region in Siberia and Alaska. *Geophys. Res. Lett.* **40**, 6165–6170 (2013).
- Sepulveda-Jauregui, A. et al. Eutrophication exacerbates the impact of climate warming on lake methane emission. *STOTEN* **636**, 411-419 (2018).
- Tan, Z., & Zhuang, Q. Methane emissions from pan-Arctic lakes during the 21st century: An analysis with process-based models of lake evolution and biogeochemistry. *J. Geophys. Res. Biogeosci.* **120**, 2641–2653 (2015).

- Yoshikawa, K. & Hinzman, L. D. Shrinking thermokarst ponds and groundwater dynamics in discontinuous permafrost near Council, Alaska. *Permafr. Periglac. Process.* **14**, 151-160 (2003).
- van Huissteden, J. et al. Methane emissions from permafrost thaw lakes limited by lake drainage. *Nat. Clim. Change* **1**, 119–123 (2011).
- van Vuuren D. P. et al. The representative concentration pathways: An overview. *Clim. Change* **109**, 5–31 (2011).
- Velichko, A. A. et al. Climate changes in East Europe and Siberia at the Late glacial–holocene transition. *Quat. Int.* **91**, 75–99 (2002).
- Walter, K. M., Edwards, M. E., Grosse, G., Zimov, S. A., & Chapin, F. S. Thermokarst lakes as a source of atmospheric methane during the last deglaciation. *Science* **318**, 633-636 (2007).
- Walter Anthony, K. M., Anthony, P., Grosse, G., & Chanton, J. Geologic methane seeps along boundaries of arctic permafrost thaw and melting glaciers. *Nat. Geosci.* **5**, 419-426 (2012).
- Walter Anthony, K. M. et al. A shift of thermokarst lakes from carbon sources to sinks during the Holocene epoch. *Nature* **511**, 452–456 (2014).
- Walter Anthony, K. M. et al. Methane emission proportional to permafrost carbon thawed in Arctic lakes since the 1950s. *Nat. Geosci.* **9**, 679-682 (2016).
- Whalen, S. C. & Reeburgh, W. S. Moisture and temperature sensitivity of CH₄ oxidation in boreal soils. *Soil Biol. Biochem.* **28**, 1271–1281 (1996).
- Yvon-Durocher, G. et al. Methane fluxes show consistent temperature dependence across microbial to ecosystem scales. *Nature* **507**, 488-491 (2014).

REVIEWERS' COMMENTS:

Reviewer #1 (Remarks to the Author):

The authors have adequately addressed the questions I had regarding the balance of lake losses and lake gains. The changes in manuscript text and addition of new supplemental information helps to quantify the sensitivity of the conclusions on the uncertainties regarding lake area dynamics.

I am not sure that the yellow arrows in Supplementary Figure 3 are helpful in their current state. I can examine the images and find multiple places where small areas of lake disappearance are also being missed in the detection scheme. Please consider the objectivity of that presentation.

Reviewer #2 (Remarks to the Author):

I believe that the authors have satisfactorily addressed the points that I raised in my review.

Reviewer #3 (Remarks to the Author):

The authors have thoroughly addressed the issues raised in my initial review. As far as my comments go, I am comfortable with the revisions and additions to the text, figures, and supplemental materials that the authors document in their response to comments, and I appreciate the authors' efforts to explain and justify their revisions.

July 10, 2018

Letter to the Referees

The reviewers' comments on our manuscript (NCOMMS-17-32396A), entitled "21st-century modeled permafrost carbon emissions accelerated by abrupt thaw beneath lakes," were helpful. We have responded to these comments in the revised manuscript (reviewer comments repeated here in bold) and attempted to address their remaining concerns in the following ways:

Reviewer #1 (Remarks to the Author):

The authors have adequately addressed the questions I had regarding the balance of lake losses and lake gains. The changes in manuscript text and addition of new supplemental information helps to quantify the sensitivity of the conclusions on the uncertainties regarding lake area dynamics.

I am not sure that the yellow arrows in Supplementary Figure 3 are helpful in their current state. I can examine the images and find multiple places where small areas of lake disappearance are also being missed in the detection scheme. Please consider the objectivity of that presentation.

The reviewer is correct. There are instances where disappearance of small lakes is also missed in the detection scheme. To improve objectivity and clarity, we have added a second set of arrows (magenta) indicating select examples of undetected losses of small lakes.

Reviewer #2 (Remarks to the Author):

I believe that the authors have satisfactorily addressed the points that I raised in my review.

Reviewer #3 (Remarks to the Author):

The authors have thoroughly addressed the issues raised in my initial review. As far as my comments go, I am comfortable with the revisions and additions to the text, figures, and supplemental materials that the authors document in their response to comments, and I appreciate the authors' efforts to explain and justify their revisions.

We thank the reviewers for their helpful remarks, which have caused us to improve the paper. We hope that the revised manuscript will now be found suitable for publication in Nature Communications.